# Comprehensive Survey of OCT-Based Disorders Diagnosis: From Feature Extraction Methods to Robust Security Frameworks

**DOI:** 10.3390/bioengineering12090914

**Published:** 2025-08-25

**Authors:** Alex Liew, Sos Agaian

**Affiliations:** 1Graduate Center, City University of New York, New York, NY 10016, USA; 2College of Staten Island, City University of New York, New York, NY 10314, USA; sos.agaian@csi.cuny.edu

**Keywords:** optical coherence tomography (OCT), hand-crafted features, deep learning models, adversarial attacks, robustness in medical imaging, security in AI model, glaucoma detection, diabetic retinopathy, clinical decision support systems

## Abstract

Optical coherence tomography (OCT) is a leading imaging technique for diagnosing retinal disorders such as age-related macular degeneration and diabetic retinopathy. Its ability to detect structural changes, especially in the optic nerve head, has made it vital for early diagnosis and monitoring. This paper surveys techniques for ocular disease prediction using OCT, focusing on both hand-crafted and deep learning-based feature extractors. While the field has seen rapid growth, a detailed comparative analysis of these methods has been lacking. We address this by reviewing research from the past 20 years, evaluating methods based on accuracy, sensitivity, specificity, and computational cost. Key diseases examined include glaucoma, diabetic retinopathy, cataracts, amblyopia, and macular degeneration. We also assess public OCT datasets widely used in model development. A unique contribution of this paper is the exploration of adversarial attacks targeting OCT-based diagnostic systems and the vulnerabilities of different feature extraction techniques. We propose a practical, robust defense strategy that integrates with existing models and outperforms current solutions. Our findings emphasize the value of combining classical and deep learning methods with strong defenses to enhance the security and reliability of OCT-based diagnostics, and we offer guidance for future research and clinical integration.

## 1. Introduction

### 1.1. Optical Coherence Tomography

Optical Coherence Tomography (OCT) is a non-invasive imaging technology essential to the field of ophthalmology. Developed in the early 1990s, OCT utilizes light waves to capture high-resolution, cross-sectional images of the retina, the light-sensitive tissue at the back of the eye. This non-invasive technology makes it beneficial because it does not require contact with the eye, making it suitable for sensitive patients or those who need frequent assessments. Furthermore, the widespread availability of OCT has made it a standard tool in clinical settings. This allows clinicians to observe the retina’s layers in detail, enabling them to detect and monitor a range of ocular diseases [1,2,3]. These observations allow for the visualization of changes in the retina that might signify early disease stages. This is significant in diagnosing conditions such as glaucoma, where early detection can prevent the progression of vision loss. Moreover, OCT plays a vital role in monitoring the progression of diseases like age-related macular degeneration, diabetic retinopathy, and other eye conditions [4,5,6,7].

Age-related Macular Degeneration (AMD) comes in two forms: dry and wet. Dry AMD is the common type and develops when parts of the macula, a small area in the center of the retina that ensures sharp vision, get thinner with age, and tiny clumps of protein called drusen grow. This causes a gradual loss of central vision. Wet AMD, also referred to as Choroidal Neovascularization (CNV), is less common but more severe and occurs when new, abnormal blood vessels grow under the retina, which can leak blood and fluids. This leakage can cause rapid damage to the macula, leading to more severe and rapid vision loss than dry AMD. Diabetic retinopathy (DR) occurs in people who have diabetes. High blood sugar levels cause damage to the blood vessels in the retina. These vessels can swell and leak, or they can close, blocking blood flow. These changes can cause central and peripheral vision loss over time. Diabetic macular edema (DME) is a subset of diabetic retinopathy. Similar to DR, high blood sugar levels damage the small blood vessels in the retina, leading them to leak fluid or bleed. When this fluid accumulates in the macula, it causes swelling, and the vision becomes blurred. DME is a significant cause of vision loss in people with diabetes [8,9,10,11].

Other ill conditions include a macular hole (MH) and Central Serous Retinopathy (CSR). A macular hole is a small break in the macula, which leads to blurring and distortion of central vision. These holes can develop from the natural shrinking of the vitreous gel that fills the eye or from injuries or other eye diseases. CSR is a condition where fluid builds up under the retina, creating a detachment that specifically affects the macula, leading to distorted and blurred vision. The condition is often stress-related and is more common in men than women. CSR usually resolves on its own, but severe cases might require treatment to prevent lasting damage to the retina. The OCT images above in Figure 1 display visualizations of the ocular disorders mentioned [12].

Figure 2 presents a pair of fundus and OCT scans, emphasizing the complementary relationship between these two retinal imaging methods. Fundus images provide a wide-field photograph of the retina, which highlights key features like blood vessels and the optic disc. These features are essential for diagnosing diseases such as diabetic retinopathy and glaucoma. The OCT scan (below) offers a detailed cross-sectional view of the retina, which belongs to a specific portion of the fundus image. Together, these images are crucial for an eye health assessment, as the fundus image identifies surface-level abnormalities, while the OCT scan reveals deeper structural issues like retinal thickening or fluid accumulation [13].

Another utility of OCT in clinical settings lies in the ability to provide detailed images, which enables analysis of these images through a process known as feature extraction. Feature extraction involves identifying specific attributes or changes in the OCT images that are relevant for diagnosing eye conditions.

### 1.2. Feature Extraction Techniques

In OCT image classification for ocular disorders, two main types of methods are used to analyze images: hand-crafted features and deep learning approaches, including Convolutional Neural Networks and Vision Transformers.

Hand-crafted features involve manually designed techniques where specific details of an image are selected based on what is already known about eye diseases. For example, experts might choose to focus on specific patterns or textures in the image that typically indicate a problem. This method relies heavily on the knowledge and experience of specialists to identify which features are important for diagnosis. While it can be very effective when the disease markers are well understood, it is less flexible and might not handle new or complex situations as well [15,16,17,18,19].

On the other hand, deep learning methods like CNNs and transformers automate the process of finding essential features in images. CNNs work by processing images through multiple layers, each designed to recognize different features, from simple edges to more complex shapes. This allows the network to understand the image in a structured way, layer by layer. CNNs are particularly good at handling images where recognizing localized patterns is key to making a diagnosis. Transformers, which were originally designed for processing text, have been adapted to work with images. They look at the entire image at once, rather than piece by piece. This helps them understand the broader context and relationships within the image, which can be beneficial in complex diagnostic scenarios where the overall structure and layout of the image elements are essential [20,21,22,23,24,25,26,27,28,29,30,31,32,33,34,35,36,37,38,39,40,41,42,43,44,45,46,47,48,49].

Both CNNs and transformers learn from examples rather than being programmed with specific rules about what to look for. They need a lot of data to learn effectively and can sometimes act like “black boxes,” making it hard to understand how they have reached their conclusions. The choice between using hand-crafted features or deep learning approaches depends on factors like the availability of data, how decisions are made, and the required level of accuracy. Understanding these algorithms’ reliance on data highlights the importance of OCT datasets. These datasets are crucial for training and testing these models, determining their effectiveness and accuracy [47,48,49,50,51,52,53,54,55,56,57,58,59,60,61,62,63,64,65,66,67].

### 1.3. Other Survey Literature on OCT

The current survey literature on OCT in ocular disorders, such as [1,2,3], primarily concentrates on specific applications of deep learning and computer vision for diagnosing and analyzing retinal diseases. These studies explore topics like automatic segmentation, the classification of retinal diseases through OCT images, and the use of deep learning for detecting conditions such as glaucoma and age-related macular degeneration. For example, surveys like [2,3] delve into the technical methodologies of image processing and the latest advancements in algorithmic approaches using OCT images. Other existing literature, such as [5,6,9], emphasizes the results of applying these advanced computational techniques without discussing the foundational feature extraction methods that still play a crucial role in scenarios where training data is limited or there are specific diagnostic features. Similarly, refs. [8,10] focus on the methodological aspect of aligning OCT images to enhance the accuracy of longitudinal studies and treatment monitoring. Table 1 compares our survey with others in the area of OCT image classification for ocular disorders, while Table 2 provides a summary of each survey.

In contrast, our survey presents a more holistic approach by bridging the gap between deep learning-based techniques and traditional hand-crafted feature extraction methods, a comparison largely absent in previous studies. While prior works such as [4,7,11] have explored deep learning approaches in various capacities, they lack discussion on the comparative effectiveness of convolutional neural networks (CNNs) and Vision Transformers (ViTs) versus traditional hand-crafted techniques. Moreover, our survey uniquely includes a comprehensive review of multiple OCT datasets, which is crucial for evaluating the generalizability of feature extraction methodologies.

Our survey also provides an extensive discussion on the datasets employed in OCT-based image analysis. The choice of datasets significantly impacts model performance, particularly in clinical settings where the variability in imaging conditions, disease prevalence, and patient demographics can affect the reliability of automated classification models. Many existing surveys rely on a limited set of public datasets, such as the datasets in [12,14,45,68,69,70,71,72,73,74], without critically evaluating their applicability to real-world clinical scenarios. In contrast, our work examines the diversity of available datasets, highlighting their strengths and limitations in terms of sample size and disease coverage. By doing so, we offer insights into how dataset selection influences model bias, generalization capability, and potential deployment in medical diagnostics.

Additionally, our survey does not merely summarize existing methods but critically evaluates their strengths, weaknesses, and applicability under different clinical and computational constraints. Unlike existing studies that primarily focus on retrospective analysis, our work also identifies key gaps in current research and suggests new directions, particularly in areas such as adversarial attacks on OCT image classification and the integration of Large Language Models (LLMs) into ocular disease diagnostics. These aspects have been largely overlooked in previous surveys, making our study a valuable contribution that extends beyond conventional literature reviews.

By addressing the intersection of deep learning and traditional feature extraction, our survey provides a comprehensive and balanced perspective, offering insights into the current capabilities and future directions of OCT image feature extraction technologies. This comparative analysis not only enhances understanding but also guides future research in a way that no other existing survey has attempted, making it a unique and essential reference for researchers in this domain.

This survey has the following contributions.

Provides a systematic review of the existing methods of feature extraction from OCT images, categorizing them into hand-crafted and deep learning-based approaches:
Evaluates these methods against various performance metrics, accuracy, precision, sensitivity, specificity, and F1 score.Evaluates and highlights the evolution from using hand-crafted features to using deep learning techniques like CNNs and transformers in enhancing feature extraction from OCT images.Assesses the impact of dataset choice on the performance of feature extraction methods.Explores the emerging field of adversarial conditions in medical imaging, particularly in OCT, to propose future directions for research that could lead to more robust, accurate, and clinically relevant feature extraction technologies.

This survey has the following sections. The “Review of OCT Datasets” section presents commonly used datasets in OCT classification. “Hand-Crafted Feature Extraction Techniques” describes recent feature-engineering techniques in OCT ocular disease classifications. The “Deep Learning Approaches” section describes neural network architectures for OCT ocular disorder detections using CNNs and transformers. “Comparative Analysis” compares the performance of hand-crafted features, CNNs, and transformers using data from various datasets. “Future Works” discusses the potential of adversarial samples to test and improve the robustness of OCT classification models. “Discussion” analyzes the findings from the comparative and dataset review sections. “Conclusion” recaps the major insights of the paper.

## 2. Review of OCT Datasets for Ocular Disorder Classification

### 2.1. OCT Datasets Details

As the OCT technology has advanced, there’s been a growing need for OCT datasets. These collections of eye images are crucial for training and testing the accuracy of models designed to spot eye problems. These models are used in deep learning to analyze images. Having a variety of high-quality OCT datasets is key to making these models as effective as possible. In this review, we will look at different OCT datasets used for identifying eye diseases. We will go over what makes each dataset unique and how they help improve the technology used in diagnosing eye conditions.

The first dataset, referred to as Dataset 1, includes volumetric scans from 45 patients, divided into three groups: 15 normal patients, 15 with dry age-related macular degeneration (AMD), and 15 with diabetic macular edema (DME). All SD-OCT volumes were collected using Spectralis SD-OCT equipment (Heidelberg Engineering Inc., Heidelberg, Germany) at Duke University, Harvard University, and the University of Michigan [68]. The second dataset, referred to as Dataset 2, comes from the Noor Eye Hospital dataset cited in the references. It includes 148 SD-OCT volumes, of which 48 are age-related macular degeneration (AMD), 50 are diabetic macular edema (DME), and 50 are normal volumes. These were captured using the Heidelberg SD-OCT imaging system at Noor Eye Hospital in Tehran (NEH). Each volume contains between 19 and 61 B-scans, with each B-scan having a resolution of 3.5 μm and the overall scan dimensions being 8.9 by 7.4 mm^2^ [45].

Creating a dataset with classes for normal, diabetic macular edema (DME), and age-related macular degeneration (AMD) is beneficial because it covers two common and significant causes of vision impairment. Normal images help the model understand what a healthy retina looks like. DME images teach models to recognize the swelling caused by fluid accumulation from damaged blood vessels in diabetes. AMD images show changes in the retina due to aging, including drusen and other abnormalities. Datasets 1 and 2 are effective for general screening tools and simplify the training process by focusing on broader categories of eye health issues.

Dataset 3 was also obtained using the Heidelberg SD-OCT imaging system at Noor Eye Hospital (NEH) and is available on the Mendeley database website as referenced in [69]. It initially included 16,822 OCT images, covering 120 volumes of Normal images, 160 volumes of Drusen, and 161 volumes of CNV (choroidal neovascularization). This dataset configuration aims at changes related to AMD. Drusen are the early indicators of AMD, and separating them into their own class allows for early detection of the disease before it potentially progresses to more severe stages, such as CNV. This setup is useful for specialists focused on monitoring and treating AMD, allowing for early intervention strategies and careful monitoring of disease progression.

Dataset 4 is a publicly available dataset known as the UCSD Dataset [70]. This dataset contains 108,312 OCT images in the training set and 1000 images in the test set. Within the training dataset, 37,206 images are CNV, 11,349 images are DME, 8617 images are Drusen, and 51,140 images are normal. A trimmed-down version is also employed in some of the literature. The trimmed-down version has the following class-count: 37,455 are CNV, 11,598 are DME, 8866 are drusens, and 26,565 are normal, with a total of 84,484 OCT images. This dataset includes both wet and dry AMD, which necessitate distinct treatment strategies. The inclusion of a separate CNV class enables the model to distinguish between these forms of AMD, while also identifying diabetic-related abnormalities and normal retinal conditions.

Dataset 5 has 384 thickness maps of the total retina from individual subjects, where 269 are subjects with intermediate AMDs and 115 subjects are free of any ocular diseases [71]. These volumetric rectangular scans were obtained from Bioptigen Inc., Research Triangle Park, NC, USA, which was approved by the institutional review boards of Devers Eye Institute, Duke Eye Center, Emory Eye Center, and National Eye Institute. A dataset with only normal and intermediate AMD OCT images narrows the focus of the diagnostic tool. It is a simpler dataset that enhances the model’s ability to detect stages of AMD, particularly the intermediate stage, which is often difficult to diagnose.

Dataset 6 consists of 24,000 images and is divided equally into eight different categories: AMD, CNV, DME, MH, DR, CSR, and healthy subjects [12]. This dataset allows for very precise diagnosis and is valuable in specialized care. For example, distinguishing between different types of AMD or recognizing characteristics of less common conditions like CSR can enable more targeted interventions. However, this model requires learning from a larger volume of data, which distinguishes differences between more categories. Similar to Dataset 6, Dataset 7 includes four classes, which are normal macula, macular edema, macular hole, and AMD [72]. Dataset 7 consists of 326 macular spectral-domain OCT scans collected from 136 subjects, encompassing a total of 193 eyes. The scans have an original resolution of either 200 × 200 × 1024 or 512 × 128 × 1024 in a 6 × 6 × 2 mm volume (width, height, and depth). The UPMC Eye Center, Eye and Ear Institute, Ophthalmology and Visual Science Research Center, Department of Ophthalmology developed this dataset. In a comparable dataset, [53] dataset 7*, the Eye Center at Renmin Hospital of Wuhan University gathered 4076 OCT images of DM patients, centered on the fovea, between 2016 and 2022. These images were obtained using an OCT device (Optovue RTVue, Optovue, Fremont, CA, USA). This dataset allows models to learn the progression of diabetic eye disease and related macular damage in detail. This helps early detection and guides personalized treatment plans based on severity.

Dataset 8 was developed by the Singapore Eye Research Institute (SERI) and was collected using the CIRRUS SD-OCT device from Carl Zeiss Meditec, Inc., located in Dublin, CA, USA. This dataset includes 32 OCT volumes, divided into 16 cases of diabetic macular edema (DME) and 16 normal cases. Each volume comprises 128 B-scans, with a resolution of 512 × 1024 pixels. All SD-OCT images were reviewed and assessed by trained graders who classified them as either normal or DME cases based on the evaluation of retinal thickening, hard exudates, intraretinal cystoid space formation, and subretinal fluid. Dataset 9 was obtained using a raster scan protocol with a 2 mm scan length, featuring a resolution of 512 × 1024 pixels. These images were captured with a Cirrus HD-OCT machine (Carl Zeiss Meditec, Inc., Dublin, CA, USA) at Sankara Nethralaya (SN) Eye Hospital in Chennai, India. For each volumetric scan, an experienced clinical optometrist (MKP) selected a fovea-centered image. Dataset 9 comprises 102 images of macular holes (MH), 55 images of age-related macular degeneration (AMD), 107 images of diabetic retinopathy (DR), and 206 normal retinal images.

Another dataset, D10, circular OCT B-scan images, collected using the swept-source OCT device (DRI-OCT, Topcon, Inc., Tokyo, Japan), focuses on a 3.4mm diameter circle centered on the optic disc and is available in various sizes. This dataset consists of 1395 samples (697 glaucoma and 698 non-glaucoma) from 641 participants, involving a total of 1015 eyes, with 135 eyes having follow-up data. Visual field tests and OCT images are provided for all participants. The dataset categorizes samples into early, moderate, and advanced stages, with 447, 140, and 110 samples, respectively. OD (right eye) samples include 201 in the early stage, 82 in the moderate stage, and 56 in the advanced stage. OS (left eye) samples include 246 in the early stage, 58 in the moderate stage, and 54 in the advanced stage [14]. Table 3 provides a summary of the information for each dataset.

After reviewing the diverse datasets used in OCT-based ocular disease prediction, we now turn our attention to significant challenges such as dataset bias, imaging heterogeneity, and domain shift.

### 2.2. Dataset Bias, Imaging Heterogeneity, and Domain Shift

While reviewing OCT datasets for ocular disease classification, it is essential to consider three key factors that influence model development and generalization: dataset bias, imaging heterogeneity, and domain shift.

***Dataset bias*** is frequently apparent in several datasets due to inherent class imbalance, which can lead to models over-prioritizing majority classes. For instance, Dataset 5 includes 269 intermediate AMD volume images but only 115 normal volume images. Such an imbalance may cause models to overfit towards the AMD class, potentially leading to inflated performance metrics that do not reflect true generalization. Similarly, Dataset 7 covers four disease categories, normal macular, macular edema, macular hole, and AMD, but the number of samples per class varies notably (e.g., 316 normal vs. 261 edema vs. 297 holes). This uneven distribution could significantly affect balanced model training and impact the model’s ability to classify minority classes accurately. In contrast, Dataset 6 stands out by providing an evenly distributed collection of 3000 images per class across eight categories, thereby supporting more stable and robust multi-class learning by mitigating the risk of class imbalance bias.

***Imaging heterogeneity*** arises from inherent differences in OCT device types, image acquisition protocols, and institutional practices. These variations can introduce inconsistencies that pose significant challenges for model generalization. For example, Dataset 1 is composed of images from multiple clinical sites, including Duke, Harvard, and Michigan. At the same time, Datasets 4 and 4* originate from distinct centers like UC San Diego and Guangzhou Women and Children’s Medical Center. Such multi-site data collection can introduce variations in resolution, image texture, signal-to-noise ratio, and illumination. These inconsistencies can potentially impact a model’s ability to generalize effectively across different imaging platforms or clinical settings, necessitating robust model designs or explicit harmonization techniques.

***Domain shift*** refers to the often-observed performance degradation when a model trained on a specific source dataset is applied to a target dataset with differing characteristics. This is a critical concern in medical imaging, where data sources often vary. For instance, a model trained predominantly on Dataset 6, compiled from institutions such as Devers, Duke, Emory, and NEI, may underperform when evaluated on Dataset 10, which includes data from Zhongshan Ophthalmic Center and Sun Yat-sen University. This degradation is typically due to variations in imaging equipment (e.g., different OCT manufacturers or models), patient demographics, disease prevalence, and even imaging technician practices between the centers. Aggregated datasets such as D3 and D6, which deliberately combine samples from multiple centers, help mitigate domain shift by exposing models to a broader range of “domains” during training. Conversely, smaller, single-source datasets like D8 (from SERI) may be more prone to overfitting domain-specific features, making models trained on them less generalizable to external clinical environments.

Having explored the critical challenges posed by dataset bias, imaging heterogeneity, and domain shift, we now shift our focus to how to analyze this complex data effectively. This brings us to a discussion of the main techniques for extracting useful information from OCT images: hand-crafted features and deep learning.

## 3. Hand-Crafted Feature Extraction Techniques

This section aims to provide a thorough overview of various hand-crafted feature extraction methods that have been developed to analyze OCT images. We explore how these techniques operate by extracting specific, predefined features from images such as texture, shape, and intensity. These predefined features are known to be indicators of ocular disorders. These features are then used to classify, segment, and analyze OCT data in the context of diagnosing conditions. Specifically, articles that will be reviewed employ techniques such as Local Binary Patterns (LBPs) and Dictionary Learning, which have been effective in extracting meaningful features from OCT images.

Local binary patterns (LBPs) are a technique used to describe the local spatial patterns and texture of an image. In the context of OCT imaging, LBP helps in identifying fine-grained patterns within the retina that may indicate early signs of diseases such as macular degeneration or diabetic retinopathy. The method works by comparing each pixel with its neighbors and encoding these relationships into a binary code, which effectively captures the texture information. The classical local binary pattern (LBP) is a texture image descriptor that emphasizes the center pixel and its neighboring pixels to encode structural texture information within an image. The generalized form of an LBP is expressed as follows:(1)$$LBP\left(I_{C}\right)=\sum_{i\epsilon R}f\left(i\right)s\left(I_{i}-I_{C}\right)$$
where *I_C_* represents the center pixel, *I_i_* represents the adjacent surrounding pixels, and *f*(*i*) = 2*i*, *i* = 0, …, 7, with *R* representing a region defined by the kernel size. The function *s*(*I_i_* – *I_C_*) assigns a value of 1 if the difference between the surrounding pixel and the center pixel is greater than or equal to zero (T is set to zero); otherwise, it assigns a value of 0. Each kernel is placed over a pixel (*I_C_*) and compared to its surrounding neighbors (*I_i_*) using the mentioned function. A binary sequence is generated based on these comparisons, and each sequence is assigned a corresponding decimal weight of *f*(*i*). The following are works developed in the past ten years or more.

A machine learning method has been developed to classify OCT images for three retina-related diseases: macular hole (MH), age-related macular degeneration (AMD), diabetic retinopathy (DR), and normal (NO) OCT images. This method employs LBPs to extract features from the images and utilizes a classifier that operates on the Random Forest technique to differentiate between the disease states and normal conditions [15]. A low-complexity feature vector connection method, known as slice-sum, has been introduced to reduce the computational load required by the SVM classifier. The detector employs only the LBP and SVM classifier, which helps minimize the hardware resources needed for processing [16]. A method has been developed to extract global descriptors from the 2D feature image for LBPs and from the 3D volume OCT image. As a result, the global-LBP mapping technique will extract *d* feature elements [17].

The method involves a standard classification process that includes initial preprocessing steps to eliminate noise and flatten each B-Scan. It utilizes features like Histogram of Oriented Gradients (HOG) and LBPs, which are extracted and then merged to form various feature vectors. These vectors are then input into a linear Support Vector Machine (SVM) classifier for further analysis [18]. A method of local texture descriptor known as Multi-Kernels Wiener Local Binary Patterns (MKW-LBP) is used for the classification of eye diseases such as age-related macular degeneration, diabetic macular edema, and normal eyes. The accuracy of this descriptor is optimized using classification techniques such as support vector machines (SVMs), Adaboost, and Random Forest. The experimental evaluations demonstrate that MKW-LBP achieves superior diagnostic and recognition performance when compared to recent developments in texture descriptors [19]. Similar methods develop local texture descriptor algorithms, Multi-Size Kernels ξcho-Weighted Median Patterns (MSKξMP) and Alpha Mean Local Binary Patterns (AMT-LBPs), to avoid speckle noise and classify eye diseases like DME and AMD. The methods also employ Singular Value Decomposition to achieve optimal accuracy with the SVM and Random Forest classification techniques [75,76].

The method presents an automatic detection method that combines discrete wavelet transform (DWT) image decomposition, local binary pattern (LBP)-based texture feature extraction, and multi-instance learning (MIL). LBP is chosen for its ability to handle low contrast and low-quality images, minimizing the interference from the image itself on the detection method. DWT image decomposition supplies high-frequency components rich in details for extracting LBP texture features, removing redundant information unnecessary for diagnosing CSCR in the raw image [77]. Other hand-crafted feature extractors are also employed and are discussed below. Another method is a machine learning approach that utilizes global image descriptors derived from a multi-scale spatial pyramid. Local features are dimension-reduced local binary pattern histograms, which encode texture and shape information in retinal OCT images and their edge maps. This representation works at multiple spatial scales and granularities, resulting in robust performance. Two-class support vector machine classifiers are used to identify the presence of normal macula and three specific pathologies. Additionally, to distinguish subtypes within a pathology, we build a classifier to differentiate full-thickness holes from pseudo-holes within the macular hole category [72].

A two-feature-labeling method for the 3D OCT volume: the slice-chain labeling method and the slice-threshold labeling method. These methods are evaluated using the SVM [78]. The approach utilizes retinal features like retinal thickness, individual retinal layer thickness, and volumes of pathologies such as drusen and hyper-reflective intra-retinal spots. The approach automatically extracts ten clinically important retinal features from segmented SD-OCT images for classification. The effectiveness of these features is evaluated using several classification methods, including Random Forest [79]. Another approach, a contrast enhancement-based adaptive denoising, is used to eliminate speckle noise. Pixel grouping and iterative elimination, based on typical layer intensities and positions, are used to identify the RPE layer. Randomization techniques, followed by polynomial fitting and drusen removal, are then applied to estimate a baseline. Classification is determined by comparing the drusen height to the baseline [80]. A method for automated detection of retinal diseases in eyes uses Histogram of Oriented Gradients (HOG) descriptors and support vector machines (SVMs) to classify each image within a spectral domain (SD)-OCT volume as either normal, containing dry AMD, or containing DME [68].

Finally, the last two methods are based on dictionary learning. An approach utilizing HOG features of pyramid images combined with three different dictionary learning methods—Separating the Particularity and the Commonality dictionary learning (COPAR), Fisher Discrimination Dictionary Learning (FDDL), and Low-Rank Shared Dictionary Learning (LRSDL)—was investigated to achieve the highest classification accuracy of OCT images [81]. Another approach proposes a general framework for distinguishing normal OCT images from DME and AMD scans using sparse coding and dictionary learning. This includes a preprocessing and alignment technique for the retina to address the shortcomings of previous methods, which struggle to classify datasets with severely distorted retina regions. Additionally, sparse coding and structured preprocessing (SP) are employed, along with an SVM for classification [82]. Table 4 shows the results of the hand-crafted-feature extractor work discussed.

## 4. Deep Learning Approaches

This section aims to provide a thorough overview of applications of CNNs in OCT image classifications. Various CNN architectures have been explored to enhance the feature extraction and accuracy. Typically, in CNNs, the core operation is the convolution applied across multiple layers. The convolution at the l-th layer is mathematically expressed as:(2)$$h_{i,j}^{\left(l\right)}=\sum_{m}\sum_{n}W_{mn}^{\left(l\right)}X_{\left(i+m\right)\left(j+n\right)}^{l-1}+b^{\left(l\right)}$$
where *X*^(*l*)^ is the input feature map from the previous layer (or the raw image if it is the first layer, *W*^(*l*)^ is the convolution filter at layer *l*, *b*^(*l*)^ is the bias term at layer l, and hij^(l)^ is the output feature map at position (*i*,*j*) for layer *l*. A non-linear activation function, such as ReLU, is applied to the result of the convolution:(3)$$a_{ij}^{\left(l\right)}=ReLU\left(h_{ij}^{\left(l\right)}\right)=\max\left(0,h_{ij}^{\left(l\right)}\right)$$
where this operation is repeated across multiple convolutional layers, allowing the network to extract more features. After the convolutional layers, the pooling layers reduce the spatial dimensions:(4)$$p_{ij}^{\left(l\right)}=\underset{\left(m,n\right)\in \mathrm{Window}}{\max}\left(a_{\left(i+m\right)\left(j+n\right)}^{\left(l\right)}\right)$$ The pooling window reduces the resolution of the feature map.

Next, augmentation CNNs leverage data augmentation techniques to expand the training dataset artificially, improving model robustness and performance. Standard augmentation techniques include rotation, flipping, and cropping. Image augmentation is often used to create diverse training samples, reduce overfitting, and improve the model’s generalization ability. The papers reviewed will include techniques beyond standard methods. CNNs with specialized augmentation using Generative Adversarial Networks (GANs) aim to augment the training data by generating synthetic but realistic images. This augmentation improves the network’s ability to generalize, especially when the training data is scarce or imbalanced. GAN-based augmentation can be formulated as:(5)$$X_{aug}=G\left(z\right)$$
where *G*(*z*) is the generator network of the GAN, which produces synthetic images from a noise vector *z*, and *X_aug_* is the generated augmented image. By training the CNN on both real and GAN-generated images, the model becomes more robust to variations and improves generalization.

Additionally, regular CNNs enhanced with residual units and inception units have shown significant promise. Residual units help in mitigating the vanishing gradient problem, allowing for deeper networks that can learn more complex features. Residual units in CNNs help to mitigate the vanishing gradient problem, allowing the network to train deeper architectures. The residual block is defined as:(6)$$y^{\left(l\right)}=F\left(X^{\left(l\right)},W^{\left(l\right)}\right)+X^{\left(l\right)}$$
where the equation represents the transformations (convolutions, activations) applied to the input *X*^(*l*)^ at layer *l*. *X*^(*l*)^ is added directly to the output, forming a shortcut connection. Inception units, which consist of multiple convolutions with different kernel sizes, enable the network to capture a hierarchy of features by processing the input in parallel. Together, these diverse CNN architectures form the backbone of state-of-the-art deep learning approaches for ocular disease prediction from OCT images. Inception units process the input using multiple convolution filters with different sizes, enabling the network to capture features at multiple scales in parallel. The inception unit can be formulated as:(7)$$y=\left[f_{1\times 1}\left(X\right),f_{3\times 3}\left(X\right),f_{5\times 5}\left(X\right),Pooling\left(X\right)\right]$$
where *f*_1×1_(*X*), *f*_3×3_(*X*), and *f*_3×3_(*X*) represent convolutions with different filter sizes, Pooling X is an additional pooling operation that captures larger-scale information. By combining different filter sizes, the inception unit allows the network to capture both fine and coarse details from the input image.

Segmentation-based attention CNNs incorporate attention mechanisms that focus on the most relevant regions of the OCT images, thus improving the detection of subtle pathological features. This approach often combines segmentation tasks with the primary classification task, ensuring that the network pays attention to critical areas while learning. The attention mechanism generates an attention map *A*(*X*), which weighs different regions of the feature map based on their relevance:(8)$$A\left(X\right)=\sigma \left(W_{a}*X\right)$$
where *σ* is the generic function, typically a sigmoid, that generates the attention weights, *W_a_* is the attention filter, and * denotes convolution. The attention map is applied to the feature map to emphasize the most relevant areas:(9)$$X_{att}=A\left(X\right)\cdot X$$
where *X_att_* is the attention-weighted feature map that focuses the network’s attention on critical regions of the OCT image.

Ensemble CNNs are another prominent strategy, where multiple CNN models are trained independently, and their predictions are combined to produce a final output. Let *f_i_*(*X*) represent the prediction of the *i*-th CNN in the ensemble. The final prediction y from the ensemble is computed as an average of all individual model outputs:(10)$$y=\frac{1}{N}\sum_{i=1}^{N}f_{i}\left(X\right)$$
where *N* is the number of CNN models in the ensemble, and *f_i_*(*X*) is the prediction from the *i*-th model. This method employs the strengths of different models, leading to improved predictive performance and reduced variance.

Multi-scale CNNs, on the other hand, process OCT images at various scales, capturing features at different levels of detail. This multi-resolution approach enables the network to identify both coarse and fine-grained features, which is particularly useful in detecting a wide range of ocular diseases. The multi-scale processing is defined as:(11)$$y=\left[f_{R_{1}}\left(X\right),f_{R_{2}}\left(X\right)\ldots ,\;f_{R_{K}}\left(X\right)\right]$$
where *f_R_*_1_(*X*), *f_R_*_2_(*X*), …, *f_RK_*(*X*) represent the convolutions applied to the input image *X* at lower (R1) to higher (RK) resolutions. The outputs from different scales are then combined, allowing the network to analyze features across multiple resolutions in parallel. Figure 3 shows the different types of CNN structures discussed above. At the same time, Table 5 summarizes the different CNN architectures shown in Figure 3 for OCT eye disease classification, along with merits and scenarios where each model may perform best.

### 4.1. CNNs

This section explores standard and advanced CNN techniques, including residual and inception units, which improve feature learning and network depth, enabling the prediction of ocular diseases from OCT images.

A hybrid Retinal Fine-Tuned Convolutional Neural Network (R-FTCNN) has been proposed for detecting retinal diseases such as diabetic macular edema, drusen, and choroidal neovascularization from OCT images. This study employs the R-FTCNN architecture alongside principal component analysis (PCA) as part of its methodology. PCA was used to transform the fully connected layers of the R-FTCNN into principal components, and the Softmax function was applied to these principal components to create a new classification model [20]. The approach introduces a deep learning framework that leverages dual guidance between two tasks. First, a Complementary Mask Guided Convolutional Neural Network (CM-CNN) is employed to classify OCT B-scans, distinguishing between normal scans and those with drusen or CNV. This classification is guided by masks generated from an auxiliary segmentation task. Second, a Class Activation Map Guided UNet (CAM-UNet) is used for segmenting drusen and CNV lesions, utilizing the CAM output from the CM-CNN [21]. Another work presents a framework for the automated detection of retinal disorders utilizing transfer learning. The model operates in three phases: deep fused and multilevel feature extraction using 18 pre-trained networks and tent maximal pooling, feature selection with ReliefF, and classification with an optimized classifier [22].

Another technique involves removing the final layers from the pre-trained Inception V3 model and utilizing the remaining portion as a fixed feature extractor. The extracted features are then fed into a CNN designed to learn the shifts in the feature space [23]. An automated CNN architecture, AOCT-Net, has been proposed for a multiclass classification system based on OCT. This system, incorporating a Softmax classifier, is designed to classify five types of retinal diseases: AMD, CNV, DME, drusen, and typical cases [24]. Another method, an iterative fusion convolutional neural network (IFCNN), adopts an iterative fusion strategy, which combines features from the current convolutional layer with those from all previous layers in the network. This approach enables the joint utilization of features from different convolutional layers, leading to accurate classification of OCT images [25]. Another work introduced OCT Deep Net2 for classifying optical coherence tomography images. This study performed a four-class disease classification, with OCT Deep Net2 being an extension of OCT Deep Net1, expanding from 30 to 50 layers. OCT Deep Net2 is a dense architecture featuring three recurrent modules [26]. Another model, based on a capsule network, is designed to enhance classification accuracy. Capsules, which are groups of neurons representing different properties of the same object, use vectors to learn positional relations between features in images. This reportedly offers higher generalization performance than traditional CNNs for small affine transformations of training data, thus requiring far fewer training samples [27].

A dictionary learning method to reduce image size leverages DAISY descriptors and Improved Fisher kernels to extract OCT image features. Similar to traditional downsampling methods, the approach functions as a form of intelligent downsampling, effectively reducing image size while preserving essential information [28]. Another work introduced two methods for detecting retinal abnormalities from OCT images. The first method, termed S-DDL, offers a solution to the vanishing gradient problem in DDL and reduces training time. The second method utilizes the Wavelet Scattering Transform (WST), which incorporates predefined filters in network layers. The two methods are compared [29]. Another method proposed a weakly supervised deep learning framework with uncertainty estimation to classify macula-related diseases from OCT images, utilizing only volume-level labels. First, a convolutional neural network (CNN)-based instance-level classifier is iteratively refined through our proposed uncertainty-driven deep multiple instance learning (MIL) scheme. Then, a classifier can detect suspicious abnormal instances and create deep embeddings for those instances. Second, a recurrent neural network (RNN) uses features from those instances to make final predictions [30]. Another work proposed a two-stage approach for retinal OCT volume classification, which consists of (1) volumetric feature extraction and (2) diagnostic classification. This approach utilizes a wavelet-based CNN (WCNN) feature learning subsystem in the feature extraction stage. The WCNN includes a spatial-frequency decomposition layer (SFD-layer) in the first hidden layer, which serves as feature learning in retinal OCT B-scans [31]. Table 6 presents the performance metrics for each of the CNN methods using the datasets discussed in this section.

### 4.2. CNN with Attention

This section reviews papers on segmentation-based attention CNNs, which enhance OCT image analysis by using attention mechanisms to focus on critical regions, improving subtle pathological feature detection and integrating segmentation with classification tasks for better learning.

A study introduced a method called the lesion-aware CNN (LACNN) approach for retinal OCT image classification, utilizing retinal lesions within OCT images to guide the CNN for more accurate classification. The LACNN focuses on local lesion-related regions in the OCT images using a lesion detection network to create a soft attention map from the entire OCT image [32]. Another approach integrates a dual-attention mechanism at multiple levels of a pre-trained deep convolutional neural network (CNN). It enhances focused learning by incorporating both multilevel feature-based attention, which targets salient coarse features, and a self-attention mechanism, which focuses on higher entropy regions of the finer features [33]. Another method proposes a deep architecture based on a perturbed composite attention mechanism, incorporating two attention modules: Multilevel Perturbed Spatial Attention (MPSA) and Multidimensional Attention (MDA) for macular optical coherence tomography (OCT) image classification. MPSA enhances the salient regions of input images and the features from intermediate network layers by adding positive perturbations to the attention layers. Conversely, MDA encodes the normalized interdependency of spatial information across various channels of the extracted feature maps. This perturbed composite attention enables the architecture to extract diagnostic features at different levels of feature representation [34].

A one-stage attention-based method was proposed for retinal OCT image classification and segmentation using bounding box-level supervision. Specifically, the classification network generates a heatmap using Gradient-Weighted Class Activation Mapping and incorporates the proposed attention block. Transformation consistency is employed to ensure that the predicted heatmap remains consistent for the same input after image transformation [35]. Another study presents an efficient Global Attention Block (GAB) for feed-forward convolutional neural networks (CNNs). The GAB creates an attention map across three dimensions for any intermediate feature map and then computes adaptive feature weights by multiplying the attention map with the input feature map. This GAB can be integrated into any CNN [36]. Another work proposes a B-scan attentive convolutional neural network (BACNN). BACNN is a CNN-based feature extraction module that is employed to extract spatial feature representations from the B-scans. Subsequently, a self-attention module aggregates these features according to their clinical relevance, resulting in a discriminative high-level feature vector for reliable diagnosis [37].

### 4.3. CNN Ensembles and Multi-Scale

This section reviews papers on ensemble CNNs and multi-scale approaches. Ensemble CNNs involve independently training multiple CNN models and combining their predictions to produce a final output. Multi-scale approaches process OCT images at various scales, capturing features at different levels of detail.

One approach proposes a 6G-enabled IoMT method that minimizes human involvement in medical facilities while delivering rapid diagnostic results. This method utilizes transfer learning to extract features from medical images and is enhanced by feature selection by employing operators from the Hunger Games search [38]. Another work proposes a framework that leverages deep ensemble learning, wherein the input fundus and OCT scans are processed through a deep CNN. The deep CNN first recognizes and processes the scans, which are then fed into a second layer of the CNN model to extract essential feature descriptors from both images. These extracted descriptors are concatenated and passed to a supervised hybrid classifier, such as support vector machines and naïve Bayes models. These classifiers are combined to achieve accurate classification [39]. Another approach involves combining features from various resolutions, leading to the following discussion: multi-scale CNNs.

A method of employing a multi-scale deep feature fusion (MDFF)-based classification approach using CNNs for reliable diagnosis is proposed. The MDFF technique captures inter-scale variations in the images, providing the classifier with discriminative information [40]. Another proposed architecture is a multi-scale and multipath CNN comprising six convolutional layers. The multi-scale convolution layer enables the network to generate local structures capturing both sparse local and detailed global structures [41]. Another paper introduces a multi-scale (CNN) architecture for the accurate diagnosis of AMD. The proposed architecture consists of a multi-scale CNN with seven convolutional layers designed to classify images as either AMD or normal. The multi-scale convolution layer allows for the generation of numerous local structures with various filter sizes [42]. Finally, another method proposes a novel multi-scale CNN with a feature pyramid network (FPN). The model leverages multi-scale receptive fields to enhance the accurate detection of retinal pathologies of varying scales in OCT images [43]. Due to the advantages of utilizing both ensemble and multi-scaling techniques, the following papers implement a combination of these approaches.

Another method proposes a multi-stage classification network based on a multi-scale (pyramidal) feature ensemble architecture. Initially, a scale-adaptive neural network generates multi-scale inputs for feature extraction and ensemble learning. Larger input sizes capture more global information, while smaller input sizes focus on local details. Subsequently, a feature pyramidal architecture is designed to extract multi-scale features, utilizing DenseNet as the backbone [44]. A similar approach presents a system based on a multi-scale convolutional mixture of experts (MCME) ensemble model. The proposed MCME modular model employs a new cost function for discriminative learning of image features by applying CNNs on multiple scales. MCME maximizes the likelihood function of the training dataset and ground truth by using a Gaussian mixture model [45]. Finally, another approach proposed a Deep Multi-Scale Fusion CNN (DMF-CNN) that encodes multi-scale disease characteristics. Specifically, multiple CNNs with different receptive fields are utilized to obtain scale-specific feature representations from the OCT images. These representations are then fused to extract cross-scale discriminative features for classification. Additionally, a joint multi-loss optimization strategy is employed to collectively learn scale-specific and cross-scale complementary information during training [46]. Table 7 presents the performance metrics for each of the specialized CNN methods discussed above.

### 4.4. CNN Augmentations

In this section, we review papers on CNN classification, focusing on how specialized augmentation enhances the model’s generalization by generating diverse training samples. One method proposes a surrogate-assisted classification method for automatically classifying retinal OCT images using convolutional neural networks (CNNs). The process involves image denoising, followed by thresholding and morphological dilation to extract masks, which are used to generate surrogate images for training the CNN model. The final prediction for a test image is determined by averaging the outputs from the CNN model on these surrogate images [47]. Another approach developed a semi-supervised classifier based on a GAN for automated diagnosis using limited labeled data. This framework includes a generator and a discriminator, where adversarial learning between the two helps create a generalizable classifier capable of predicting progressive retinal diseases such as age-related macular degeneration and diabetic macular edema [48]. Another work introduces an unsupervised framework using a GAN to achieve fast and reliable super-resolution. Adversarial learning with cycle consistency and identity mapping priors ensures the preservation of spatial correlation, color, and texture details in the generated HR images, which are then used for classification tasks [49].

### 4.5. Transformers

While CNNs and their variations have significantly advanced image processing, transformers have elevated it to new heights. Vision Transformers (ViTs), derived from the transformer architecture in Natural Language Processing (NLP), achieve outstanding benchmark results on ImageNet datasets, representing a significant leap forward in computer vision.

In a standard ViT architecture, the input image is first divided into fixed-size patches, which are then flattened and linearly projected into embeddings. Let $x_{p}\in R^{H\times W\times C}$ represent an input image of height *H*, width *W*, and *C* channels. The image is split into patches of size *P* × *P*, resulting in N = *H*^⋅^*W*/*P*^2^ patches, where each patch is a vector of $x_{p}\in R^{P^{2}C}$. These patches are linearly embedded using:(12)$$z_{0}^{i}=x_{p}^{i}\cdot E,\;i=1,2,\;\ldots ,\;N$$
where $x_{p}\in R^{{(p}^{2}C)\times D}$ is the learnable embedding matrix, and $z_{0}^{i}$ represents the patch embeddings of dimension *D*. Next, a positional encoding is added to retain spatial information:(13)$$z_{0}=\left[x_{p}^{1}E;\;x_{p}^{2}E;\ldots ;\;x_{p}^{N}E\right]+\;E_{pos}$$
where $E_{pos}\in R^{N\times D}$ is the positional encoding matrix. The sequence of patch embeddings is then fed into a standard transformer encoder, consisting of multiple layers of Multihead Self-Attention (MHSA) and feed-forward networks (FFN). For each layer l, the self-attention mechanism is computed as:(14)$$Attention\left(Q,K,V\right)=softmax\left(\frac{QK^{T}}{\sqrt{D_{k}}}\right)V$$
$Q=Z_{l-1}W_{Q}\;$, $K=Z_{l-1}W_{K}$, and $V=Z_{l-1}W_{V}$ are the query, key, and value matrices, respectively, and *D*_k_ is the dimensionality of the key. The output of the self-attention mechanism is passed through a feed-forward network:(15)$$z_{l}^{\prime }=MHSA\left(z_{l-1}\right)+z_{l-1}$$(16)$$z_{l}^{\prime }=FFN\left({z\prime }_{l}\right)+z_{l}$$

After the final transformer layer, the class token (a learnable embedding added to the input sequence) is extracted and passed to a classifier for the final prediction. The following are reviews of papers on the application of transformers to OCT images for predicting eye disorders.

An approach hybrid ConvNet–Transformer network (HCTNet) begins with a low-level feature extraction module, utilizing a residual dense block to generate features that facilitate network training. Following this, two parallel branches, one using a transformer and the other a ConvNet, are designed to capture the global and local contexts of the OCT images. Finally, a feature fusion module with an adaptive reweighting mechanism is employed to combine these global and local features for accurate OCT image categorization [50]. A method introduces an Interpretable Swin-Poly ViT network for automated retinal OCT image classification. By shifting the window partition, the Swin-Poly transformer establishes connections between adjacent non-overlapping windows from the previous layer, allowing it to model multi-scale features flexibly.

Additionally, the Swin-Poly transformer adjusts the significance of polynomial bases to refine cross-entropy, enhancing the accuracy of retinal OCT image classification [51]. Another study proposes Focused Attention, which uses iterative conditional patch resampling to generate interpretable predictions via high-resolution attribution maps, addressing the low-resolution issue of existing Transformer attribution methods. A survey involving four retinal specialists validated both the superior interpretability of Vision Transformers compared to CNN attribution maps and the relevance of Focused Attention as a lesion detector [52]. A method utilizing a Vision Transformer can more effectively capture global information through its self-attention mechanism and exhibits less bias towards local texture features. The classifier is redesigned using logits and the loss function as the logit cross-entropy function with L2 norm [53].

One paper introduces a technique called the model-based transformer (MBT). This technique leverages pre-trained models, specifically the ViT and Swin Transformer, for OCT image classification and the multi-scale ViT for OCT video classification. The proposed method represents OCT data using an approximate sparse representation technique, then estimates the optimal features for classification [54]. Another paper introduces a framework called the Structure-Oriented Transformer (SoT), designed to enhance the relationship modeling between lesions and the retina regions. A model-oriented filter highlights the entire retina structure and guides relationship construction. Then, a pre-trained ViT is employed to model the relationships among all feature patches through transfer learning.

Additionally, to optimize the use of all output tokens, a vote classifier is employed to obtain final grading results [55]. Similarly, another approach proposes an OCT Multihead Self-Attention (OMHSA) block to process OCT image information using a hybrid CNN-ViT approach. The OMHSA incorporates local information extraction into the self-attention calculation and adds local information to the transformer model. A neural network architecture, named OCTFormer, is employed by repeatedly stacking convolutional layers and OMHSA blocks at each stage [56]. Another study introduces a hybrid SqueezeNet-Vision Transformer (SViT) model, which leverages the strengths of both SqueezeNet and Vision Transformer (ViT). This model captures both local and global features of OCT images, enabling more accurate classification while maintaining lower computational complexity [57].

Another article proposes a Deep Relation Transformer (DRT) for glaucoma diagnosis by combining OCT and Vision Field (VF) information. This model introduces a deep reasoning mechanism to explore implicit pairwise relations between OCT and VF data, both globally and regionally. Also, three successive modules are developed to extract and collect information for glaucoma diagnosis: the Global Relation Module, the Guided Regional Relation Module, and the Interaction Transformer Module [14]. A fusion model called “Conv-ViT” employs transfer learning-based CNN models, such as Inception-V3 and ResNet-50, to process texture information by calculating the correlation of nearby pixels. Additionally, a Vision Transformer model is integrated to process shape-based features by determining the correlation between long-distance pixels [58]. Another article proposes a ViT-based cross-modal multi-contrast network for integrating color fundus photographs (CFP) and optical coherence tomography (OCT) images. The approach employs multi-contrast learning to extract features from cross-modal data for diagnosis. Subsequently, a channel fusion head captures the semantically shared information across different modalities and the similarity features among patients within the same category [59].

Another set of architectures involves the following. One approach proposes a deep learning model based on the Swin Transformer V2 to diagnose fundus diseases swiftly and accurately. This method leverages the calculation of self-attention within local windows to reduce computational complexity and enhance classification efficiency. Additionally, the PolyLoss function was introduced to boost the model’s accuracy further [60]. A method called lesion-localization convolution transformer (LLCT) uses customized feature maps generated by a convolutional neural network (CNN) as the input sequence for a self-attention network. This design leverages CNN’s ability to extract image features and the transformer’s capacity to consider global context and dynamic attention. Part of the model undergoes backpropagation to calculate the gradient as a weight parameter, which is then multiplied and summed with the global features generated during the forward propagation process to locate the lesion [61] accurately. A stitching approach to find an optimal model by combining two MedViT family models is proposed. This method, known as stitchable neural networks, is an efficient architecture search algorithm. It creates a candidate model in the search space by inserting a linear layer between each pair of stitchable layers, with each layer in the pair being selected from one of the input models [62]. Finally, in another study, a deep learning framework was developed that utilizes the diagnostic potential of 3D OCT imaging for automated glaucoma detection. The framework integrates a pre-trained Vision Transformer on retinal data for slice-wise feature extraction and a bidirectional Gated Recurrent Unit (GRU) to capture inter-slice spatial dependencies. This dual-component approach allows for an analysis of both local details and global structural integrity [63]. Table 8 presents the performance metrics for each of the transformer methods discussed above.

The following are short works presented at conferences that are slight modifications of ViT. One work proposed a CAD method using a base Vision Transformer to analyze OCT images and distinguish between AMD, DME, and normal eyes [64]. Another approach aimed to develop a deep learning algorithm to distinguish between drusen and the double-layer sign (DLS) based on cross-sectional structural OCT B-scans, using a Vision Transformer (ViT) model trained on eye images [65]. Another conference proposes an end-to-end Transformer-based framework designed to classify volumetric data of varying lengths efficiently. By randomizing the input volume-wise resolution (number of slices) during training, we enhance the learnable positional embedding’s ability to adapt to each volume slice [66]. Finally, another ViT is proposed using a symmetrical cross-entropy loss function, which can minimize the effect of noise on the training set and prevent overfitting [67].

## 5. Comparative Analysis

### 5.1. Accuracy Comparisons Based on Datasets

In this section, we discuss the performance of hand-crafted features, CNNs, and transformer models in predicting ocular disorders using OCT data across a series of well-established datasets. Figure 4 presents an overview of the various techniques discussed with their corresponding classification accuracies. Focusing first on Dataset D4, which is crucial for distinguishing between the dry and wet forms of age-related macular degeneration (AMD) and recognizing diabetic-related changes and normal conditions, we observe a range of techniques with varying effectiveness. For example, the multi-contrast network achieves a high accuracy of 99.75%, indicating its robustness in handling the complexities of D4. Similarly, models like HCTNet and Conv-ViT also perform well, with accuracies of 91.56% and 94.46%, respectively. These high accuracies suggest that these techniques are well-suited for applications requiring precise differentiation between similar conditions, such as distinguishing dry AMD from wet AMD, which is critical for appropriate treatment planning.

In the context of D2 and D5, which cater to broader screening processes and more specialized monitoring for AMD, several techniques stand out. For instance, the LBP Slice + Sum and SVM technique applied to D5 achieves an accuracy of 87.3%, which is particularly useful for detecting intermediate stages of AMD, which is a challenging task for many models. D2, which focuses on general screening, sees strong performance from CNN-based methods such as MPSA (99.62%) and D2FPN-DenseNet (90.9%). These techniques are valuable in clinical settings where quick and reliable screening is essential for early intervention. On the other hand, D3, designed for AMD monitoring, benefits from techniques like Interpretable Swin-Poly, which offers an accuracy of 99.8%. This high level of accuracy is crucial for specialists who require reliable tools to monitor disease progression and adjust treatment plans accordingly.

For the remaining datasets (D1, D6, D7, D8, and D9), the figure highlights a diverse set of techniques tailored to specific clinical needs. D1, for example, is well-served by traditional CNN approaches like R-FTCNN and CNN iterative ReliefF, both achieving perfect accuracies of 100%, making them highly effective for general screening purposes. D6, which involves distinguishing between different types of AMD and less common conditions like CSR, benefits from advanced models like Stitched Tiny MedViTs with an accuracy of 98.6%, offering doctors a reliable tool for targeted interventions. Meanwhile, D7*, which includes a variety of diabetic macular edema (DME) stages, finds MSK-EMP with an accuracy of 96.62% particularly suitable, aiding in precise diagnosis and treatment decisions. Finally, for D9, which covers a broader range of conditions, techniques like ViT with Logit Loss Function (87.3%) and Interpretable Swin-Poly (97.31%) offer substantial accuracy, providing clinicians with dependable tools for diagnosing diverse retinal conditions. Each technique’s suitability is closely tied to its ability to support doctors in making informed decisions, whether through accurate screening, detailed monitoring, or distinguishing between subtle variations in retinal diseases.

### 5.2. Task Relevance, Data Dependency, Interpretability, or Robustness

This section introduces a structured comparative framework to evaluate the OCT classification models surveyed in our study. The selected dimensions, task relevance, data dependency, interpretability, and robustness, are critical for assessing not only performance in controlled settings but also real-world clinical applicability.

Task relevance assesses the model’s effectiveness in addressing specific diagnostic objectives, such as detecting age-related macular degeneration (AMD) or differentiating between multiple ocular diseases.Data dependency examines the scale and complexity of the datasets required for training, which directly impacts model generalizability in data-constrained settings.Interpretability considers the extent to which the model’s decision-making process can be understood and trusted by clinicians, fostering confidence and facilitating clinical integration.Lastly, robustness evaluates a model’s resilience to variations in input data, including noise, artifacts, and rare disease presentations, ensuring reliable performance in diverse clinical environments.

Together, these dimensions offer a comprehensive framework for comparing the strengths and limitations of OCT-based classification models.

Task Relevance: The relevance of a model to specific diagnoses depends on its design and performance across different disease-specific datasets. For example, models trained on Dataset D6, which covers eight distinct classes including AMD, choroidal neovascularization (CNV), diabetic macular edema (DME), macular hole (MH), diabetic retinopathy (DR), and Central Serous Retinopathy (CSR), achieved strong multi-class performance. Specifically, MSK-EMP scored 99.76% and CM CNN achieved 98.3% accuracy, demonstrating their capability to handle complex diagnostic tasks.

Dataset D1, although smaller in size, presents a three-class problem (DME, AMD, and normal) and still supports high-performing models such as ViT and MSPE, both reaching 99.69% accuracy. These results indicate that while high accuracy can be achieved with limited class numbers, broader task relevance is better captured by models trained on diverse datasets like D4*, D6, and D3, which span a wider range of retaining disorders and better reflect real-world diagnostic challenges.

Data Dependency: The size and variety of the training data strongly influences the effectiveness of OCT-based classification models. Models like the multi-contrast network (99.75%) and Interpretable Swin-Poly (99.8%) achieved high performance when trained on large and diverse datasets such as D4* and D6. These datasets include over 80,000 to 100,000 labeled 2D OCT images covering multiple disease types, and this extensive dataset helped these models generalize well across different diagnostic categories.

On the other hand, models trained on smaller datasets like D5 (384 images) or D1 (45 images) also reported high accuracy (e.g., MSPE and ViT both with 99.69%). However, their success tends to be limited to simpler classification tasks with fewer classes. These findings suggest that while strong results can be achieved with smaller datasets for specific, limited tasks, acquiring diverse and extensive data is crucial for developing robust models capable of multi-disease classification in real-world clinical settings.

Interpretability is an essential quality in medical imaging AI, as clinicians must understand and trust the model’s decision-making process. A subset of reviewed models explicitly incorporates interpretability mechanisms, such as attention maps or saliency region visualizations. For example, the “Interpretable Swin-Poly” model and “LBP Slice-Sum & SVM” offer insights into how features contribute to classification outcomes, which can be invaluable for clinical validation and understanding. In contrast, highly complex models like deep CNN ensembles or transformer-based architectures often act as “black boxes,” offering limited inherent interpretability unless augmented with additional tools (e.g., LIME, SHAP). Future research should prioritize developing intrinsic interpretability for these complex models or integrating robust post hoc explanation methods to enhance clinician trust and adoption.

Model robustness was evaluated based on performance across noisy or complex datasets that simulate real-world variability. Models such as MSK-EMP (99.76%), trained on D1, D2, and D3, retained high accuracy, suggesting a degree of resilience to the variations present in these datasets. Additionally, hand-crafted models like AMT-LBP and MKY-LBP, each achieving over 98.5% accuracy, demonstrated robustness to Gaussian and Salt and Pepper noise.

However, the evaluation of models under adversarial attacks or explicitly perturbed inputs remains limited in the current literature. The use of varied and complex datasets like D6, which includes underrepresented conditions such as CSR and macular holes, can serve as indirect indicators of robustness by exposing models to a broader range of clinical presentations. Future studies should focus on more rigorous evaluations of robustness, including stress testing with diverse noise models, adversarial examples, and external validation datasets to ensure reliable performance in clinical practice.

## 6. Future Work

Future research in ocular disorder prediction using optical coherence tomography (OCT) should critically address and focus on two pivotal and interrelated challenges: enhancing the robustness of deep learning models against adversarial attacks in medical imaging and effectively integrating Large Language Models (LLMs) into the diagnostic process to leverage their advanced reasoning capabilities.

### 6.1. Enhancing Robustness Against Adversarial Attacks in Medical Imaging

As the field of medical imaging continues to evolve, a significant emerging concern is the vulnerability of deep learning models to adversarial samples. These are intentionally crafted data designed to mislead models into making incorrect predictions. In OCT images, even slight, imperceptible perturbations can lead to catastrophic misclassifications, posing a severe risk in clinical settings where a misdiagnosis can have serious implications for patient outcomes. The growing recognition of these vulnerabilities has prompted researchers to explore robust defense mechanisms and adversarial training strategies to improve model resilience [19,83,84,85,86,87,88,89,90,91,92,93,94].

While existing works (e.g., MKW-LBP [19], OS-LBP for skin cancer [83], CQ-MPP for WCE images [84]) have demonstrated some robustness against specific types of noise or degradation, and studies like [85,86] explore various attack methods (FGSM, BIM, PGD), several significant challenges remain for real-world applications:Lack of Comprehensive Threat Models for OCT: Most current research on adversarial attacks focuses on generic image attacks. Future work needs to develop OCT-specific adversarial attack models that account for the unique characteristics of OCT data (e.g., speckle noise, depth information, volumetric nature, specific retinal layers) and their clinical relevance. The subtle, structured nature of retinal pathologies requires highly targeted and clinically plausible perturbations.Real-world Applicability of Defenses: While adversarial training [87,88,89], feature fusion (MEFF [91]), multi-view classification with dissonance measures [92], and knowledge-guided training [93] show promise, their effectiveness, computational overhead, and generalizability in diverse, high-stakes clinical OCT environments are yet to be fully validated. A key challenge is developing defense mechanisms that are both effective against a wide range of *unknown* attacks and computationally feasible for rapid clinical deployment without hindering diagnostic speed.Robustness to Diverse Perturbations and Imaging Artifacts: Existing studies often focus on a limited set of synthetic noise types (Gaussian, Salt and Pepper, contrast degradation [83,84]) or specific attack methods (FGSM, PGD [85,86]). Future research must develop models resilient not only to a broader spectrum of adversarial perturbations but also to realistic, naturally occurring imaging artifacts (e.g., motion blur, saturation, poor fixation, operator-dependent variations) which can mimic adversarial effects in real-world clinical OCT acquisition.Quantifying and Certifying Robustness: A significant challenge is to establish rigorous and standardized metrics, benchmarks, and formal verification frameworks for quantifying and, ideally, certifying the adversarial robustness of OCT diagnostic models. This is critical for building clinician trust, facilitating regulatory approval, and ensuring patient safety.Impact of 3D Volumetric OCT: While [94] briefly mentions 3D MRI, the complex challenges of adversarial attacks on 3D volumetric OCT data, which contain significantly more information and structural dependencies than 2D images, are largely unexplored. Perturbing 3D data subtly across slices while maintaining clinical plausibility is a major technical hurdle.Cross-Domain Robustness: Models trained on specific OCT devices or patient populations may be vulnerable to domain shift. A challenge is to develop methods that ensure adversarial robustness *across* different OCT scanners, imaging protocols, and demographic groups, without requiring extensive re-training.Adversarial Training Data Generation: Generating sufficiently diverse and clinically relevant adversarial examples for robust training, especially for rare diseases or subtle pathological changes, remains a challenge. Synthetic data generation and advanced data augmentation techniques could play a role but need careful validation. Table 9 summarizes the techniques discussed above.

### 6.2. Incorporating Large Language Models (LLMs) in Ophthalmic Diagnostics

There is growing interest in leveraging the advanced capabilities of Large Language Models (LLMs) like GPT, BERT [95], and Llama [96] in medical diagnostics. Traditionally used for natural language processing, LLMs are now being explored for their potential to interpret complex medical data, provide clinical reasoning, and support diagnostic decisions. By combining LLMs with medical imaging, such as OCT scans, the goal is to create advanced diagnostic systems that can analyze both visual and text-based information, potentially leading to more accurate and comprehensive predictions.

Recent studies have begun investigating LLM applications in ophthalmology, such as the DeepDR-LLM system for diabetic retinopathy [97] and the evaluation of GPT-4V for various ophthalmic conditions [98]. Work on using ChatGPT for fundus fluorescein angiography reports [99] and its general potential in ophthalmology [100] highlights promising avenues. However, significant challenges and unexplored opportunities remain:Multimodal Integration Challenges and Architectures: A key challenge lies in effectively and efficiently integrating visual features extracted from OCT images (which are often high-dimensional and complex) with the textual reasoning capabilities of LLMs. This requires developing sophisticated, robust, and often computationally expensive architectures that can seamlessly fuse image embeddings with patient history, symptoms, clinical notes, and potentially even genetic data, enabling holistic diagnostic reasoning beyond current capabilities.Interpretability and Explainability of LLM-driven Diagnostics: While LLMs can generate rich textual explanations, ensuring their reasoning aligns with medical best practices, is medically sound, and is truly interpretable by clinicians is crucial. The challenge is to develop methods where LLMs can not only make predictions but also provide transparent, evidence-based, and auditable justifications that clinicians can trust, verify, and use for their own decision-making process. This includes pinpointing specific image regions or textual elements contributing to a diagnosis.Handling Medical Nuances, Context, and “Hallucinations”: LLMs, trained on vast general text corpora, may struggle with the subtle nuances, specific terminology, rare disease presentations, and context-dependent reasoning inherent in complex medical cases. Ensuring LLMs do not “hallucinate” clinically inaccurate information or provide misleading medical advice remains a critical safety and ethical challenge, particularly in a field where slight misinterpretations can have severe consequences.Data Privacy and Security for Sensitive PHI: The use of LLMs, especially large-scale cloud-based models, raises significant data privacy and security concerns when handling sensitive patient health information (PHI). Future research must explore secure, privacy-preserving LLM integration methods, potentially involving privacy-preserving fine-tuning, on-device processing, or federated learning approaches, to comply with strict medical regulations (e.g., HIPAA, GDPR).Validation and Benchmarking for Multimodal Systems: Robust and clinically meaningful validation frameworks are needed to rigorously evaluate LLM-integrated diagnostic systems on diverse, real-world OCT datasets and patient cohorts. This requires developing new benchmarks for multimodal reasoning tasks that go beyond simple classification accuracy, assessing their reliability, clinical utility, and the consistency of their explanations across various ocular disorders and patient demographics.Integration into Clinical Workflow: A significant practical challenge is the seamless and user-friendly integration of LLM-powered diagnostic tools into existing clinical workflows without overburdening clinicians or disrupting established practices. This includes intuitive interfaces and efficient information flow.Computational Resources and Accessibility: Deploying and fine-tuning large LLMs, especially for specialized medical tasks, often requires substantial computational resources. Making these powerful tools accessible and scalable for clinics and research institutions with limited resources is a practical challenge.Ethical, Regulatory, and Accountability Considerations: The deployment of LLM-driven diagnostic tools necessitates careful consideration of complex ethical implications (e.g., bias amplification, patient autonomy), navigating stringent regulatory guidelines, and establishing clear accountability frameworks in cases of misdiagnosis or adverse outcomes. This involves defining the roles of AI and human clinicians.

### 6.3. Proposals for Future Research Directions

Building upon the identified challenges, we present the following specific directions for future research:Robustness to Diverse Noise and Degradation: Future research in OCT disorder prediction must prioritize the inclusion of OCT images corrupted by various types of noise (e.g., Gaussian, Salt and Pepper, uniform, speckle, Rayleigh noise, as shown in Figure 5) and realistic clinical artifacts. Incorporating these into training and validation datasets is crucial for rigorously assessing the robustness of deep learning models under less-than-ideal, real-world clinical conditions. Furthermore, LLMs could be explored to assist in identifying and characterizing different types of noise, enabling automated and adaptive preprocessing techniques. This approach could complement traditional noise reduction strategies by providing more precise noise recognition and guiding targeted denoising, thereby leading to enhanced model performance.Adversarial Testing and Defense Frameworks for OCT: Another promising direction involves the systematic incorporation of adversarial testing into OCT feature extraction and classification frameworks. Methods and frameworks designed to test the resilience of OCT models against a broad spectrum of adversarial attacks, including those specific to 3D OCT, are essential. This includes developing preprocessing techniques specifically tailored to detect and remove adversarial samples. These techniques might involve advanced adversarial training, where models are explicitly exposed to and learn to defend against diverse adversarial examples during training, or utilizing robust denoising autoencoders to filter out subtle perturbations before inference. By proactively addressing the challenge of adversarial robustness, future OCT-based AI models can be made significantly more reliable, maintaining high accuracy and sensitivity even under adverse clinical conditions.Data Dependency and Availability: While CNNs excel with large datasets, their reliance on extensive, high-quality annotated datasets remains a significant limitation in clinical settings, where data acquisition is often costly, time-consuming, and prone to scarcity.Dataset Bias, Imaging Heterogeneity, and Domain Shift: Inherent biases within datasets (e.g., class imbalance), variations introduced by different OCT device types and acquisition protocols (imaging heterogeneity), and performance degradation when models are deployed in new clinical environments (domain shift) profoundly impact model generalizability and reliability. Addressing these factors is paramount for clinical translation.Interpretability and Trust: Despite high accuracy, the “black-box” nature of many deep learning models makes it challenging for clinicians to understand how a diagnosis is reached. Enhancing model interpretability is vital for fostering trust and facilitating clinical adoption.Integration of Multimodal Information and Reasoning: The effective incorporation of advanced models like Large Language Models (LLMs) alongside image analysis presents both an opportunity and a significant challenge. Successfully fusing visual OCT data with textual clinical information (patient history, symptoms) and enabling complex clinical reasoning is crucial for comprehensive diagnostic support, yet it introduces complexities in model architecture, training, and validation.Explainable AI (XAI) for LLM-integrated Multimodal OCT Systems: Future work should focus on developing advanced XAI techniques tailored explicitly for multimodal LLM-integrated OCT diagnostic systems. This involves not only generating comprehensive textual explanations of diagnostic reasoning but also creating intuitive visual saliency maps from the OCT images, directly linked to the LLM’s decision-making pathways. The goal is to provide clinicians with transparent, verifiable, and actionable diagnostic insights, bridging the gap between AI predictions and clinical understanding.Federated Learning and Privacy-Preserving AI for OCT: Given data privacy concerns, exploring federated learning approaches for training robust OCT diagnostic models across multiple institutions without centralizing sensitive patient data is crucial. This extends to developing privacy-preserving techniques for LLM integration, such as differential privacy or secure multiparty computation, to ensure that patient data remains protected while leveraging collective data insights.Longitudinal OCT Data Analysis and Prognosis with LLMs: Expanding beyond single-time-point diagnosis, future research should explore how LLMs can integrate sequential OCT scans and patient history to predict disease progression, treatment response, and long-term prognosis. This involves challenges in handling time-series data and complex patient narratives.

By focusing on these multifaceted challenges and proposed research directions, the field can advance towards developing more reliable, secure, interpretable, and clinically actionable AI systems for ocular disorder diagnosis and management using OCT.

## 7. Conclusions

The application of optical coherence tomography (OCT) has made substantial progress in diagnosing ocular disorders, proving itself an essential tool for the early detection and monitoring of various retinal conditions. Our comparative analysis of hand-crafted feature extraction methods and deep learning techniques highlights clear distinctions in their strengths and weaknesses. While traditional feature extraction relies heavily on domain knowledge and expert intervention, making it rigid and less adaptable to data variations, deep learning approaches, particularly convolutional neural networks (CNNs), demonstrate a superior ability to learn relevant features from raw data automatically. This inherent robustness to data variability, further demonstrated by the evaluation of various CNN architectures incorporating attention mechanisms and multi-scale feature extraction, underscores the immense potential of deep learning in improving the accurate prediction of ocular disorders.

Additionally, the integration of deep learning techniques with OCT imaging holds immense potential for revolutionizing the early detection and precise diagnosis of ocular disorders. The ability to automate feature extraction from OCT images not only reduces the need for manual intervention but also significantly accelerates the diagnostic process.

Future research must rigorously focus on overcoming the current limitations, particularly by enhancing model robustness against adversarial attacks and various noise types through advanced techniques such as adversarial training and robust data augmentation. Furthermore, addressing dataset biases, imaging heterogeneity, and effectively leveraging the multimodal reasoning capabilities of LLMs will be critical.

By tackling these multifaceted challenges, the synergistic use of OCT and deep learning can significantly improve ocular disorder diagnosis, ultimately leading to more timely and effective patient outcomes. These advancements are expected to:Validate the potential of automating OCT image interpretation, enhancing ocular diagnostics, improving patient outcomes, and optimizing clinical decision-making and healthcare practices.Highlight the critical role of OCT in refining patient-specific therapeutic approaches, guiding clinicians toward increasingly personalized post-infarction therapy.Lead to the optimization of antithrombotic, lipid-lowering, and, when necessary, anti-inflammatory therapies, further improving patient clinical management.

## Figures and Tables

**Figure 1 bioengineering-12-00914-f001:**
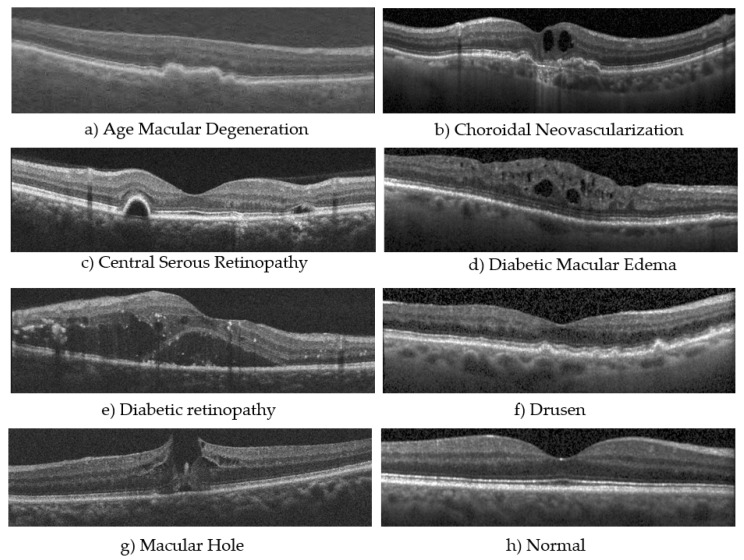
OCT images of various ocular disorders: (top row) (**a**) AMD, (**b**) CNV, (**c**) CSR, (**d**) DME; (**e**) DR, (**f**) Drusen, (**g**) MH, and (**h**) Normal. These images are taken from [12].

**Figure 2 bioengineering-12-00914-f002:**
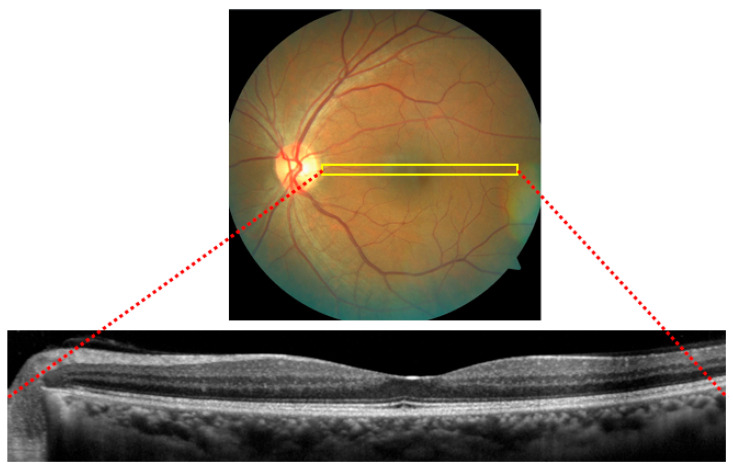
The corresponding relationship between OCT and Fundus image taken from [14].

**Figure 3 bioengineering-12-00914-f003:**
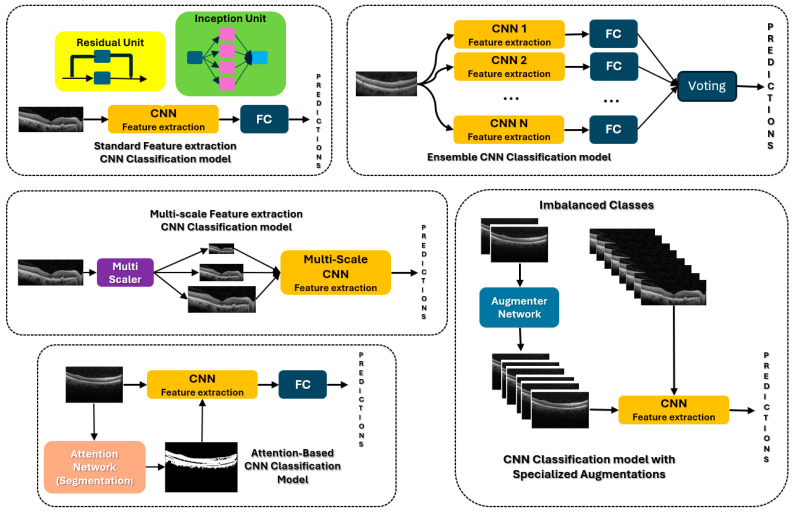
Different types of CNN structures.

**Figure 4 bioengineering-12-00914-f004:**
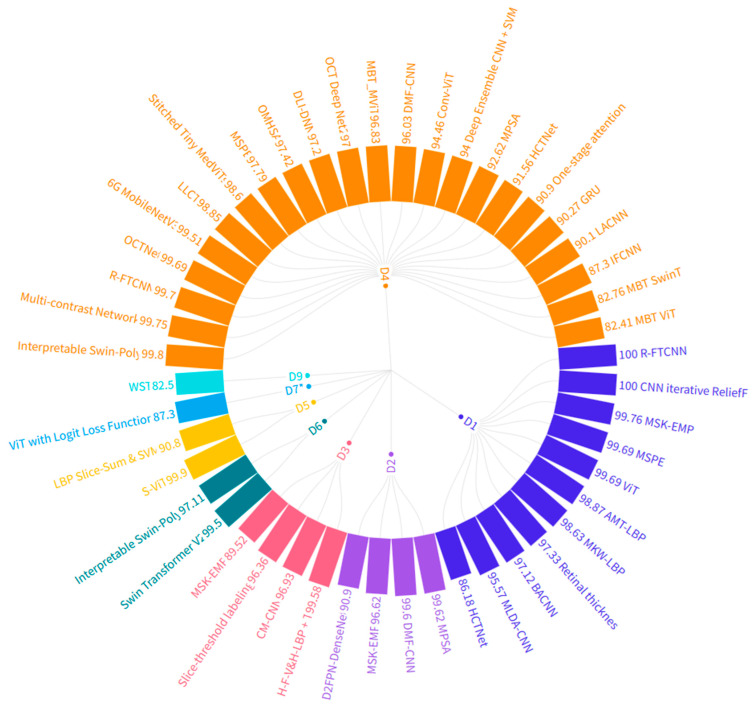
A radial bar plot comparing the performance of various techniques used in OCT ocular disorder detection across multiple datasets, indicated as D1 through D9. Each bar represents a specific technique, with the length of the bar corresponding to the classification accuracy (%) achieved by that technique.

**Figure 5 bioengineering-12-00914-f005:**
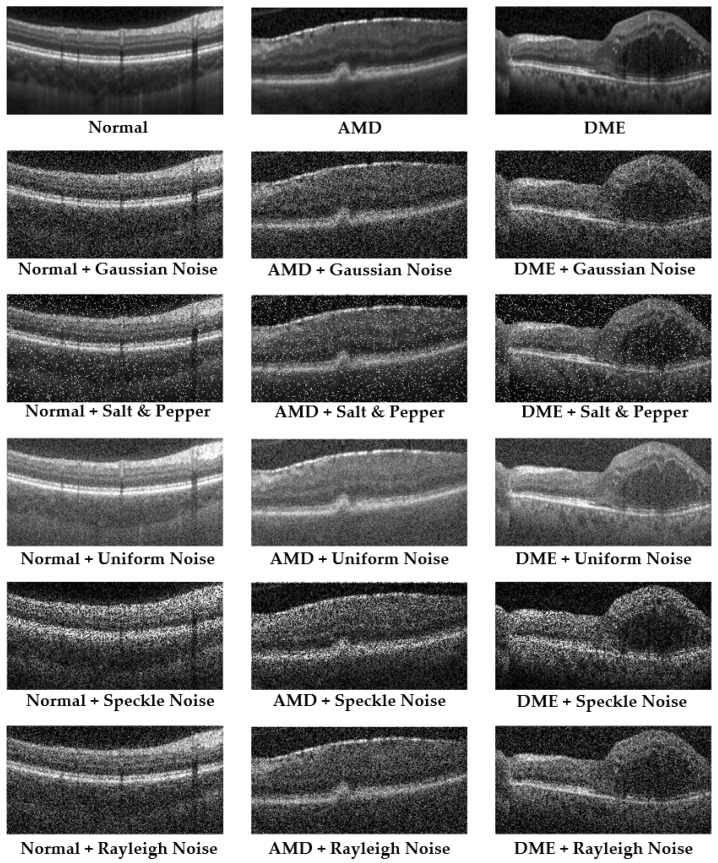
Gaussian, Salt and Pepper, uniform, speckle, and Rayleigh noise (by rows) are added to the normal, AMD, and DME (by columns), where the first column is the original. Images taken from D1.

**Table 1 bioengineering-12-00914-t001:** Our survey against others in the area of OCT image classification for ocular disorders.

Feature Comparison	[1]	[2]	[3]	[4]	[5]	[6]	[7]	[8]	[9]	[10]	[11]	[13]	OUR
Covers both DL and hand-crafted features	** *-* **	** *-* **	** *-* **	** *-* **	** *-* **	** *-* **	** *-* **	** *-* **	** *-* **	** *-* **	** *✓* **	** *-* **	** *✓* **
In-depth discussion on hand-crafted features	** *-* **	** *-* **	** *-* **	** *-* **	** *-* **	** *-* **	** *-* **	** *✓* **	** *✓* **	** *-* **	** *✓* **	** *✓* **	** *✓* **
In-depth discussion on CNNs	** *✓* **	** *✓* **	** *✓* **	** *✓* **	** *✓* **	** *✓* **	** *✓* **	** *✓* **	** *✓* **	** *✓* **	** *✓* **	** *✓* **	** *✓* **
In-depth discussion and comparison of each type of CNN	** *-* **	** *-* **	** *-* **	** *-* **	** *-* **	** *-* **	** *-* **	** *-* **	** *-* **	** *-* **	** *-* **	** *-* **	** *✓* **
In-depth discussion on Vision Transformers	** *-* **	** *-* **	** *-* **	** *-* **	** *-* **	** *-* **	** *-* **	** *-* **	** *-* **	** *-* **	** *✓* **	** *-* **	** *✓* **
In-depth comparisons between types of CNNs	** *-* **	** *-* **	** *-* **	** *-* **	** *-* **	** *-* **	** *-* **	** *-* **	** *-* **	** *-* **	** *✓* **	** *✓* **	** *✓* **
Includes comparative analysis of DL and HCF	** *-* **	** *-* **	** *-* **	** *-* **	** *-* **	** *-* **	** *-* **	** *-* **	** *-* **	** *-* **	** *-* **	** *-* **	** *✓* **
Includes in-depth discussion of ocular disorders	** *-* **	** *-* **	** *-* **	** *-* **	** *-* **	** *-* **	** *-* **	** *-* **	** *-* **	** *-* **	** *✓* **	** *✓* **	** *-* **
Discusses latest advancements	** *-* **	** *✓* **	** *✓* **	** *✓* **	** *-* **	** *-* **	** *✓* **	** *✓* **	** *✓* **	** *-* **	** *✓* **	** *✓* **	** *✓* **
Review of multiple OCT datasets	** *-* **	** *-* **	** *-* **	** *-* **	** *-* **	** *-* **	** *-* **	** *-* **	** *-* **	** *-* **	** *✓* **	** *-* **	** *✓* **
Reviews specific OCT imaging techniques	** *-* **	** *-* **	** *-* **	** *-* **	** *-* **	** *-* **	** *-* **	** *-* **	** *-* **	** *-* **	** *✓* **	** *✓* **	** *✓* **
Identifies gaps in current research	** *-* **	** *✓* **	** *✓* **	** *✓* **	** *✓* **	** *✓* **	** *✓* **	** *✓* **	** *✓* **	** *-* **	** *✓* **	** *✓* **	** *✓* **
Suggests future research into adversarial attacks	** *-* **	** *-* **	** *-* **	** *-* **	** *-* **	** *-* **	** *-* **	** *-* **	** *-* **	** *-* **	** *-* **	** *-* **	** *✓* **
Suggests future research in LLMs	** *-* **	** *-* **	** *-* **	** *-* **	** *-* **	** *-* **	** *-* **	** *-* **	** *-* **	** *-* **	** *-* **	** *-* **	** *✓* **

**Table 2 bioengineering-12-00914-t002:** A summary of each survey.

Survey	Scope of the Survey	Limitations/Gaps
[1]	Comprehensive narrative review of DL techniques for retinal layer segmentation in OCT; contrasts DL with traditional and early ML methods.	No benchmarking or comparative performance table; focuses only on segmentation
[2]	Disease-specific review linking symptoms and OCT manifestations; emphasizes CAD system design.	No quantitative comparisons or benchmarking; limited detail on DL methods
[3]	Comparative review of OCTA-based methods across applications; structured as a handbook for researchers.	Limited dataset discussion
[4]	Focused review on OCT image registration; discusses applications like speckle noise reduction and longitudinal tracking.	Limited discussion of datasets and comparative performance
[5]	Highlights performance metrics of CAD tools for AMD diagnosis; discusses early detection and telemedicine potential.	Focuses only on AMD; lacks detailed algorithmic comparison and dataset analysis
[6]	Joint analysis of OCT and Fundus-based approaches for GL and AMD; supports early detection research.	No detailed performance evaluation or dataset review
[7]	Machine learning approaches by modality and method; highlights major technical trends in both fundus and OCT domains.	Lacks a detailed comparative performance analysis
[8]	Clinical and technical features for glaucoma detection using OCT and Fundus imaging. Bridges clinical features with computational methods and highlights anatomical biomarkers for diagnosis.	Lacks quantitative comparison or algorithmic benchmarking
[9]	Deep learning applications for glaucoma detection using OCT (2D/3D/B-scans) and highlights DL’s clinical potential with OCT, covering 2D and 3D data.	Lacks specific dataset info
[10]	Systematic review of AI-based retinal screening using OCT images, which surveys numerous articles from a decade, 2013–2023, covering broad disease coverage; also comparing ML vs. DL.	Lacks dataset-level comparison.
[11]	Comparison of hand-crafted vs. deep features for OCT classification, where a quantitative comparison shows that DL outperforms hand-crafted features.	Lacks in-depth hand-crafted vs. deep features and dataset comparisons
[13]	Deep learning methods for automated ophthalmic disease diagnosis, where 99 deep learning studies are reviewed. Discusses modality-specific challenges and trends; emphasizes clinical integration and interpretability.	No benchmarking; lacks dataset-level discussion

**Table 3 bioengineering-12-00914-t003:** A summary of the information in each dataset.

Dataset	Classes and Counts	Dataset Characteristics	Institutional Source
1	15 DME volume images, 15 AMD volume images, and 15 normal volume images	Includes images of two major causes of vision impairment and a healthy baseline for comparison, which can be employed to train models to recognize diabetic swelling and age-related retinal changes, respectively.	Duke University, Harvard University, and University of Michigan
2	48 AMD volume images, 50 DME images, 50 normal images	Provides a larger collection of images than D1, representing two leading causes of vision impairment along with healthy cases, offering a strong baseline for training models to detect diabetic swelling and age-related retinal abnormalities.	Noor Eye Hospital in Tehran (NEH)
3	120 normal volume images, 160 drusen volume images, and 161 CNV volume images; 16,822 3D OCT images total	Focuses on identifying changes associated with AMD, particularly by isolating drusen as an early indicator of the disease, which monitors disease progression toward advanced stages like CNV.	Noor Eye Hospital in Tehran (NEH)
4*	37,206 CNV 2D images, 11,349 DME images 8617 drusen 2D images, 51,140 normal 2D images	Includes both wet and dry AMD, necessitating different treatment strategies, which enables training model(s) to distinguish between these forms of AMD, while identifying DME and normal retinal conditions.	University of California San Diego, Guangzhou Women and Children’s Medical Center
4	Trimmed-down version of 4* referred to as OCT2017 37,455 CNV 2D images, 11,598 DME 2D images, 8866 drusens 2D images, and 26,565 normal 2D images, total of 84,484 OCT images	This trimmed-down version of Dataset 4 requires reduced computational demand, which leads to less training time.	University of California San Diego, Guangzhou Women and Children’s Medical Center
5	269 intermediate AMD volume images and 115 normal volume images	A simplified dataset narrows the diagnostic focus and improves the model’s ability to detect the intermediate stage of AMD.	Boards of Devers Eye Institute, Duke Eye, Center, Emory Eye Center, and National Eye Institute
6	3000 AMD images, 3000 CNV images, 3000 DME images, 3000 MH images, 3000 DR images, 3000 CSR images, 24,000 total 2D OCT images	This dataset enables differentiation between various AMD types and rarer conditions like CSR and requires the model to learn from a larger volume of data to detect the eight classes.	Boards of Devers Eye Institute, Duke Eye, Center, Emory Eye Center, and National Eye Institute
7	Normal macular (316), macular edema (261), macular hole (297), AMD (284)	This dataset trains models to detect key macular diseases, each with distinct structural changes such as retinal swelling and fluid buildup in OCT images.	UPMC Eye Center, Eye and Ear Institute, Ophthalmology and Visual Science Research Center
7*	3319 OCT images total, 1254 early DME, 991 advanced DME, 672 severe DME, and 402 atrophic maculopathy	This dataset helps models learn the detailed progression of diabetic eye disease and macular damage across different severity levels.	Renmin Hospital of Wuhan University
8	16 DME volume images and 16 normal volume images	This dataset helps models to learn distinctions between healthy and diabetic macular edema cases.	Singapore Eye Research Institute (SERI)
9	Macular holes, MH (102), AMD (55), diabetic retinopathy, DR (107), and normal retinal images (206)	This dataset helps train models to detect a range of retinal conditions, including macular holes, which cause central vision loss, while DR results from diabetes-related vessel damage.	Cirrus HD-OCT machine (Carl Zeiss Meditec, Inc., Dublin, CA, USA) at Sankara Nethralaya (SN) Eye Hospital in Chennai, India
10	1395 samples (697 glaucoma and 698 non-glaucoma)	This helps improve the model’s ability to distinguish between healthy eyes and glaucoma.	Zhongshan Ophthalmic Center, Sun Yat-sen University

**Table 4 bioengineering-12-00914-t004:** List of hand-crafted methods.

Refs.	Method	Method’s Descriptions	Performance Summary
[16]	LBP Slice-Sum and SVM	Low-complexity feature vector slice-sum with SVM classifier.	^D5^Method: Accuracy (%), Sensitivity (%), LBP-RIU2: 90.80, 93.85, 87.72
[17]	3D-LBP	Global descriptors extracted from 2D feature images for LBPs and from the 3D volume OCT image. Features are fed into classifier for predictions.	^D9,V^ACC% F1% SE% SP% Global-LBP: 81.2 78.5 68.7 93.7 Local-LBP: 75.0 75.0 75.0 75.0 Local-LBP-TOP: 75.0 73.3 68.7 81.2
[18]	HOG + LBP	Histogram of Oriented Gradients (HOG) and LBP features are extracted and combined. These features are fed into a linear SVM classifier.	^D9,V^Sens, Spec, Prec, F1, Acc. HOG: 0.69 0.94 0.91 0.81 0.78 HOG + PCA: 0.75 0.87 0.85 0.80 0.81
[19]	Multi-kernel Wiener local binary patterns (MKW-LBP)	Image denoised using Wiener filter. MKW-LBP descriptor calculates the mean and variance of neighboring pixels. SVMs, Adaboost, and Random Forest are used for classification.	^D1^Kernel/Classifier: Prec. (%), Sen. (%), spec. (%), Acc (%), 3 × 3/SVM-Poly: 97.84, 97.48, 98.89, 97.86 3 × 5/SVM-Poly: 98.84, 98.59, 99.41, 98.85 5 × 5/SVM-Poly: 98.19, 98.05, 99.15, 98.33
[75]	Multi-Size Kernels Echo-Weighted Median Patterns (MSK-EMP)	Image denoised using median filter and flattened. MSKξMP is a variant of LBP that selects a weighted median pixel in a kernel and is applied to a preprocessed image. Also employs Singular Value Decomposition and Neighborhood Component Analysis-based weighted feature selection method.	Classifier: prec., sens., spec, acc ^D1^SVM-Poly: 0.9976, 0.9971, 0.9989, 0.9978 ^D2^SVM-Poly: 0.9662, 0.9663, 0.9833, 0.9669 ^D3^SVM: RBF: 0.8952, 0.8758, 0.9395, 0.8887
[76]	Alpha Mean Trim Local Binary Patterns (AMT-LBP)	Image denoised using median filter and is flattened. AMT-LBP is a variant of LBP that encodes by averaging all pixel values in a kernel and omitting highest and lowest values. SVM is employed for classification.	^D1^SVM-Poly: tr1 = 0, t_r2_ = 2 || SVM-Poly: t_r1_ = 2, t_r2_ = 0 || SVM-Poly: t_r1_ = 2, t_r2_ = 2 precision 0.9796 || 0.9846 || 0.9710 sensitivity 0.9751 || 0.9813 || 0.9654 specificity 0.9887 || 0.9920 || 0.9854 accuracy 0.9774 || 0.9836 || 0.9700 F-measure 0.9773 || 0.9829 || 0.9680 AUC 0.9740 || 0.9802 || 0.9697
[77]	H-F-V&H-LBP + T	Combines discrete wavelet transform (DWT) image decomposition, LBP-based texture feature extraction, and multi-instance learning (MIL). LBP is chosen for its ability to handle low contrast and low-quality images.	^D3,B^Acc.: 99.58%
[78]	Slice-chain labeling Slice-threshold labeling	OCT B-scans of a volume image are employed, where each slice is labeled and thresholded, which extracts features.	^D3,B^D5–Acc.: 92.50% ^D3,B^D5–Acc.: 96.36%
[79]	Retinal thickness method	The thickness of the retinal layers is measured, and each OCT image is classified according to the thickness.	^D3,B^D1–Acc.: 97.33%, Sen. 94.67%, Spec. 100%, F1: 97.22%, AUC: 0.99
[80]	RPE layer detection and baseline estimation using statistical methods	Pixel grouping/iterative elimination, guided by layer intensities, is employed to detect the RPE layer and is enhanced by randomization techniques.	^D1,V^AMD Acc: 100% Normal Acc: 93.3% DME Acc: 96.6%
[68]	Histogram of Oriented Gradients (HOG) descriptors and SVM	Noise removal using sparsity-based block matching and 3D filtering. HOG and SVM are employed for the classification of AMD and DME.	^D1,V^AMD Acc: 100% Normal Acc: 86.67% DME Acc: 100%
[81]	Dictionary learning (COPAR), (FDDL), and (LRSDL)	Image denoising, flattening the retinal curvature, cropping, extracting HOG features, and classifying using a dictionary learning approach.	^D1,V^D1–AMD Acc: 100% Normal Acc: 100% DME Acc: 95.13%
[82]	Sparse Coding Dictionary Learning	Preprocessed retina aligning and image cropping. Then, image partitioning, feature extraction, and dictionary training with sparse coding are applied to the OCT images. Linear SVM is utilized to classify images.	^D1,V^D1–AMD Acc: 100% Normal Acc: 100% DME Acc: 95.13%

^V^ Volume classification; ^B^ B-scan classification; two-class classification (normal, DME); RI: rotational invariant; U2: uniform pattern; LBP: local binary pattern; HOG: Histogram of Gradients; PCA: principal component analysis; SVM (kernel-type): support vector machine (with kernel type); t_r1_ and t_r2_: Alpha Mean Trim Factors. ^D1^D1, ^D2^D2, ^D3^D3, ^D5^D5, ^D9^D9.

**Table 5 bioengineering-12-00914-t005:** A summary of each type of CNN architecture.

CNN Structure	Key Components	Relative Merits	When It Performs Best
Standard CNN	Residual or Inception Unit → Feature Extraction → Fully Connected Layer	Simple and efficient; good for baseline performance	Suitable for balanced datasets with clear disease features
Ensemble CNN	Multiple CNNs → Independent FC Layers → Voting Mechanism	Increases robustness and reduces model variance	Effective when the dataset is diverse or noisy; improves generalization
Multi-Scale CNN	Multi-Scaler → Multiple Resolutions → Feature Extraction	Captures features at different scales; enhances spatial awareness	Best for detecting lesions of varying sizes (e.g., drusen, edema)
Attention-Based CNN	CNN → Attention Network (e.g., segmentation map) → FC	Focuses on clinically important regions; improves interpretability	Ideal when critical regions are small or when guided attention improves accuracy
CNN With Augmentation	Augmenter Network → CNN	Addresses class imbalance; improves training data diversity	Performs well when the dataset is imbalanced or small; boosts underrepresented class accuracy

**Table 6 bioengineering-12-00914-t006:** List of CNN methods.

Refs	Method	Method’s Description	Results
[20]	Hybrid Retinal Fine-Tuned Convolutional Neural Network (R-FTCNN)	R-FTCNN is employed with principal component analysis (PCA) used concurrently within this methodology. PCA converts the fully connected layers of the R-FTCNN into principal components, and the Softmax function is then applied to these components to create a new classification model.	^D1^FC1 + PCA: Acc: 1.0000, Sen.: 1.0000, Spec.: 1.0000, Prec.: 1.0000, F1: 1.0000, AUC: 1.0000 ^D4^FC1 + PCA: Acc: 0.9970, Sen.: 0.9970, Spec.: 0.9990, Prec.: 0.9970, F1: 0.9970, AUC: 0.99999 (61mil-parameters)
[21]	Complementary Mask Guided Convolutional Neural Network (CM-CNN)	CM-CNN classifies OCT B-scans by using masks generated from a segmentation task. A Class Activation Map Guided UNet (CAM-UNet) segments drusen and CNV lesions, utilizing CAM output from the CM-CNN.	^D3^AUC, Sen, Spe, Class Acc ^D3^CNV: 0.9988, 0.9960, 0.9680, 0.9773 ^D3^Drusen 0.9874, 0.9120, 0.9980, 0.9693 ^D3^Normal 0.9999, 1, 0.9880, 0.9920 ^D3^Overall Acc: 0.9693
[22]	CNN Iterative ReliefF + SVM	DeepOCT employs multilevel feature extraction using 18 pre-trained networks combined with tent maximal pooling, followed by feature selection using ReliefF.	^D1^Acc: 1.00, Pre: 1.00, F1: 1.00, Rec: 1.00, MCC: 1.00 ^4*^Acc: 0.9730, Pre: 0.9732, F1: 0.9730, Rec: 0.9730, MCC: 0.9641
[23]	Inception V3–Custom Fully Connected layers	Eliminating the final layers of a pre-trained Inception V3 model and using the remaining part as a fixed feature extractor.	^D1,V^AMD 15/15 = 100%, DME 15/15 = 100%, NOR 15/15 = 100%
[24]	AOCT-NET	Utilizes a Softmax classifier to distinguish between five retinal conditions: AMD, CNV, DME, drusen, and typical cases.	^4+5^AMD: 100%, 100%; CNV: 98.64%, 100%; DME: 99.2%, 0.96; Drusen: 97.84%, 0.92; Normal: 98.56%, 0.97
[25]	Iterative Fusion Convolutional Neural Network (IFCNN)	Employs iterative fusion for merging features from the current convolutional layer with those from all preceding layers in the network.	^D4^Sensitivity., Specificity, Accuracy Drusen 76.8 ± 7.2, 94.9 ± 1.9, 93 ± 1.7 87.3 ± 2.2; CNV 87.9 ± 4.3, 96 ± 1.7, 92.4 ± 1.3, DME 81.9 ± 6.8, 96.3 ± 2, 94.4 ± 1, Normal 92.2 ± 4.7 96 ± 1.6 94.8 ± 1.2.
[26]	IoT OCT Deep Net2	Expands from 30 to 50 layers and features a dense architecture with three recurrent modules.	^D4^Precision, Recall, F1-Score, Acc. 0.97 Normal: 0.99, 0.93, 0.96, CNV: 0.95, 0.98, 0.98, DME: 0.96, 0.99, 0.98, Drusen: 0.99, 1.00, 0.99
[27]	Capsule Network	Composed of neuron groups representing different attributes, utilizes vectors to learn positional relationships between image features.	^D4^Sensitivity, Specificity, Precision, F1 CNV: 1.0, 0.9947, 1.0, 1.0, DME: 0.992, 0.9973, 0.992, 0.992, Drusen: 0.992, 0.9973, 0.992, 0.992, Normal: 1.0, 1.0, 1.0, 1.0
[28]	Dictionary Learning Informed Deep Neural Network (DLI-DNN)	Downsampling by utilizing DAISY descriptors and Improved Fisher kernels to extract features from OCT images.	^D4^Accuracy: 97.2%, AUC: 0984, Sensitivity: 97.1%, Specificity: 99.1%
[29]	S-DDL–4 classes Wavelet Scattering Transform (WST)–5 classes	S-DDL addresses the vanishing gradient problem and shortens training time (Figure 4) -------------------------------------------- WST employs the Wavelet Scattering Transform using predefined filters within the network layers (Figure 6).	^D9^CSR-Acc: 45.5%, AMD-Acc: 64.3%, MH-Acc: 56.0%, NO-Acc: 90.7% OA: 72.0% ------------------------------------ ^D9^AMD-Acc: 9.1%, CSR-Acc: 90%, DR-Acc: 71.4%, MH-Acc: 75%, NO-Acc: 100% OA: 79.6%
[30]	Multiple Instance Learning (UD-MIL)	Employs an instance-level classifier for iteratively deep multiple instance learning, which enables the classifier. Then, a recurrent neural network (RNN) utilizes the features from those instances to make the final predictions.	^D5^Accuracy, F1, AUC μ = 0.1, 0.971 ± 0.010, 0.980 ± 0.007, 0.955 ± 0.020 μ = 0.2, 0.979 ± 0.018, 0.986 ± 0.012, 0.970 ± 0.027 μ = 0.3, 0.979 ± 0.018, 0.986 ± 0.012, 0.970 ± 0.027 μ = 0.4, 0.979 ± 0.011, 0.986 ± 0.007, 0.975 ± 0.020 μ = 0.5, 0.979 ± 0.011, 0.986 ± 0.007, 0.975 ± 0.020

^V^ Volume classification; B-scan classification; two-class classification (normal, DME). ^D1^D1, ^D3^D3, ^D4^D4, ^D5^D5, ^D9^D9.

**Table 7 bioengineering-12-00914-t007:** List of CNNs with attention, ensemble, multi-scale, and augmentation methods.

Refs.	Method	Method’s Descriptions	Results			
[32]	Lesion-Aware Convolutional Neural Network (LACNN)	LACNN concentrates on local lesion-specific regions by utilizing a lesion detection network to generate a soft attention map over the entire OCT image.	D4	Acc	Prec	
			Drusen	93.6 ± 1.4	70.0 ± 5.7	
			CNV	92.7 ± 1.5	93.5 ± 1.3	
			DME	96.6 ± 0.2	86.4 ± 1.6	
			Normal	97.4 ± 0.2	94.8 ± 1.1	
			^D4^Overall ACC: 90.1 ± 1.4, Overall Sensitivity: 86.8 ± 1.3 ^D2^Overall Sensitivity: 99.33 ± 1.49, Overall PR: 99.39 ± 1.36, F1, 99.33 ± 1.49, AUC: 99.40 ± 1.34			
[33]	Multilevel Dual-Attention-Based CNN (MLDA-CNN)	A dual-attention mechanism is applied at multiple levels of the CNN and integrates multilevel feature-based attention, emphasizing high-entropy regions within the finer features.	^D1^Acc: 95.57, Prec: 95.29, Recall: 96.04, F1: 0.996 ^D2^Acc: 99.62 (+/−0.42), Prec: 99.60 (+/−0.39), Recall: 99.62 (+/−0.42), F1: 0.996, AUC: 0.9997			
[34]	Multilevel Perturbed Spatial Attention (MPSA) and Multidimension Attention (MDA)	MPSA emphasizes key regions in input images and intermediate network layers by perturbing the attention layers. MDA captures the information across different channels of the extracted feature maps.	^D1^Acc: 100%, Prec: 100%, Recall: 100% ^D2^Acc: 99.79 (+/−0.43), Prec: 99.80 (+/−0.41), Recall: 99.78 (+/−0.43) ^D4^Acc: 92.62 (+/−1.69), Prec: 89.96 (+/−3.16), Recall: 88.53 (+/−3.26)			
[35]	One-Stage Attention-Based Framework Weakly Supervised Lesion Segmentation	One-stage attention-based classification and segmentation, where the classification network generates a heatmap through Grad-CAM and integrates the proposed attention block.	D4	Acc	SE	Spec
			CNV	93.6 ± 1.9	90.1 ± 3.8	96.5 ± 1.4
			DME	94.8 ± 1.2	86.5 ± 1.5	96.4 ± 2.1
			DRUSEN	94.6 ± 1.4	71.5 ± 4.8	96.9 ± 1.2
			NORMAL	97.1 ± 1.0	96.3 ± 1.5	98.9 ± 0.3
			^D4^OA: 90.9 ± 1.0, OS: 86.3 ± 1.8, OP: 85.5 ± 1.6			
[36]	Efficient Global Attention Block (GAB) and Inception	GAB generates an attention map across three dimensions for any intermediate feature map and computes adaptive feature weights by multiplying the attention map with the input feature map.	^D4*^Accuracy: 0.914, Recall: 0.9141, Specificity: 0.9723, F1: 0.915, AUC: 0.9914			
[37]	B-Scan Attentive Convolutional Neural Network (BACNN)	BACNN employs a self-attention module to aggregate extracted features based on their clinical significance, producing a high-level feature vector for diagnosis.	^D1^Sen: 97.76 ± 2.07, Spec: 95.61 ± 4.35, Acc: 97.12 ± 2.78,			
			D2	Sens.	Spec.	Acc.
			AMD	92.0 ± 4.4	95.0 ± 0.1	93.2 ± 2.7
			DME	100.0 ± 0.0	98.9 ± 2.4	99.3 ± 1.5
			Normal	87.8 ± 4.3	93.2 ± 2.3	92.2 ± 2.3
[38]	6G-Enabled IoMT Method–MobileNetV3	Leverages transfer learning for feature extraction and optimizes through feature selection using Hunger Games search algorithm.	D4	Acc.	Recall	Prec
			SVM	99.69	99.69	99.69
			XGB	99.38	99.38	99.4
			KNN	99.59	99.59	99.59
			RF	99.38	99.38	99.4
[39]	Deep Ensemble CNN + SVM, Naïve Bayes, Artificial Neural Network	A secondary layer within the CNN model to extract key feature descriptors, where they are subsequently concatenated and fed into a supervised hybrid classifier SVM and naïve Bayes models.	^D4^Sensivity, Specificity, Accuracy ANN: 0.96, 0.90, 0.93 || SVM: 0.94, 0.91, 0.91 NB: 0.93, 0.90, 0.91 || Ensemble: 0.97, 0.92, 0.94			
[40]	Multi-scale Deep Feature Fusion (MDFF) CNN	MDFF technique captures inter-scale variations in the images, providing the classifier with discriminative information.	D4	Sens.	Spec.	Acc.
			CNV	96.6	98.73	97.78
			DME	94.14	98.97	98.33
			DR	90.49	98.32	97.52
			NO	96.9	89.26	97.85
[41]	Multi-scale and Multipath CNN with Six Convolutional Layers	MDFF captures variations across different scales and feeds them into a classifier.		Precision	Recall	Accu.
			D1-2C	0.969	0.967	0.9666
			D2-2C	0.99	0.99	0.9897
			D4-2C	0.998	0.998	0.9978
[42]	Multi-scale CNN with Seven Convolutional Layers	The architecture consists of a multi-scale CNN with seven convolutional layers, allowing for the generation of numerous local structures with various filter sizes.	Precision Recall F1-score Accuracy AUC ^D1-2C^ 0.9687, 0.9666, 0.9666, 0.9667, 1.0000 ^D2-2C^ 0.9803, 0.9795, 0.9795, 0.9795, 0.9816 ^D4-2C^ 0.9973, 0.9973, 0.9973, 0.9973, 0.9999 ^D9-2C^ 0.9810 0.9808 0.9809 0.9808 0.9971			
[43]	Multi-scale CNN Based on the Feature Pyramid Network	Combines a feature pyramid network (FPN) and utilizes multi-scale receptive fields, providing end-to-end training.	Accuracy (%) Sensitivity (%) Specificity (%) ^D2^FPN-VGG16: 92.0 ± 1.6, 91.8 ± 1.7, 95.8 ± 0.9 ^D2^FPN-ResNet50: 90.1 ± 2.9, 89.8 ± 2.8, 94.8 ± 1.4 ^D2^FPN-DenseNet: 90.9 ± 1.4, 90.5 ± 1.9, 95.2 ± 0.7 ^D2^FPN-EfficientNetB0: 87.8 ± 1.3, 86.6 ± 1.8, 93.3 ± 0.8 ^D4^FPN-VGG16: 98.4, 100, 97.4			
[44]	Multi-scale (Pyramidal) Feature Ensemble Architecture (MSPE)	A multi-scale feature ensemble architecture employing a scale-adaptive neural network generates multi-scale inputs for feature extraction and ensemble learning.	^D1^Acc= 99.69%, Sen= 99.71%, Spec.= 99.87% ^D4^Accy = 97.79%, Sen = 95.55%, Spec. = 99.72%			
[45]	Multi-scale Convolutional Mixture of Experts (MCME) Ensemble Model	MCME model utilizes a cost function for feature learning by applying CNNs at multiple scales. Maximizing a likelihood function for the training dataset and ground truth using a Gaussian mixture model.	^D2^Precision: 99.39 ± 1.21, Recall: 99.36 ± 1.33, F1: 99.34 ± 1.34, AUC: 0.998			
[46]	Deep Multi-scale Fusion CNN (DMF-CNN)	DMF-CNN uses multiple CNNs with varying receptive fields to extract scale-specific features, which are then used to extract cross-scale features. Additionally, a joint scale-specific and cross-scale multi-loss optimization strategy is employed.	^D2^Sensitivity (%), Precision (%), F1 Score, OS, OP/OF1 AMD: 99.62 ± 0.27, 99.54 ± 0.17, 99.58 ± 0.16, 99.58 ± 0.23 DME: 99.45 ± 0.59, 99.45 ± 0.38, 99.45 ± 0.35, 99.59 ± 0.20 Normal: 99.68 ± 0.22, 99.75 ± 0.41, 99.71 ± 0.20, 99.60 ± 0.22 OA: 99.60 ± 0.21, AUC: 0.997 ± 0.002 ^D4^Sensitivity (%), Precision (%), F1 Score CNV: 97.33 ± 1.05, 97.05 ± 1.19, 97.18 ± 0.32 DME: 93.22 ± 3.22, 96.26 ± 2.17, 94.65 ± 1.09 Drusen: 89.29 ± 3.59, 87.73 ± 3.84, 88.34 ± 1.27 Normal: 97.62 ± 1.11, 97.49 ± 1.30, 97.55 ± 0.49, OS/OP/OF1/OA: 94.37 ± 1.16, 94.64 ± 0.90, 94.43 ± 0.59, 96.03 ± 0.43			
[47]	Surrogate-assisted CNN	Denoising, thresholding, and morphological dilation are performed on images to create masks, which produce surrogate images for training the CNN model.	^D1^Denoised: Acc: 95.09%, Sen. 96.39%, Spec: 93.60% ^D1^Surrogate: Acc: 95.09%, Sen. 96.39%, Spec: 93.60%			
[48]	CNN and Semi-supervised GAN		D2	Sen (%)	Spec (%)	Acc (%)
			AMD	98.38 ± 0.69	97.79 ± 0.68	97.98 ± 0.61
			DME	96.96 ± 1.32	99.23 ± 0.36	98.61 ± 0.49
			Normal	96.96 ± 0.73	99.12 ± 0.64	98.26 ± 0.67
			OS/OSp/OA: 97.43 ± 0.68, 98.71 ± 0.34, 97.43 ± 0.66			

^V^ Volume classification; ^B^ B-scan classification; ^2C^two-class classification (normal, DME). ^D1^D1, ^D2^D2, ^D3^D3, ^D4^D4, ^D4*^D4, ^D5^D5, ^D6^D6, ^D7^D7, ^D8^D8, ^D9^D9, ^D10^D10. OA: Overall Accuracy; OS: Overall Sensitivity; OP: Overall Precision; OF1: Overall F1; ^1-2C^#: binary classifications with AMD and normal classes; NB: naïve Bayes; RF: Random Forest; SVM: support vector machine.

**Table 8 bioengineering-12-00914-t008:** List of transformer methods employed.

Refs.	Method	Method’s Descriptions	Results			
[50]	Hybrid ConvNet-Transformer network (HCTNet)	HCT-Net employs feature extraction modules via a residual dense block. Next, two parallel branches, a Transformer and ConvNet, are utilized to capture both global and local contexts in the OCT images. A feature fusion module with an adaptive reweighting mechanism integrates these global and local features.	D1	Acc. (%)	Sen. (%)	Prec. (%)
			AMD	95.94	82.6	95.08
			DME	86.61	80.22	85.29
			Normal	89.81	93.39	85.22
			OA: 86.18%, OS: 85.40%, OP: 88.53%			
			D4	Acc (%)	Sen. (%)	Prec. (%)
			CNV	94.6	92.23	95.53
			DME	96.14	87.96	84.42
			Drusen	95.54	77.36	79.00
			Normal	96.84	96.73	93.5
			OA: 91.56%, OS: 88.57%, OP: 88.11%			
[51]	Interpretable Swin-Poly Transformer network	Swin-Poly transformer shifts window partitions and connects adjacent non-overlapping windows from the previous layer, allowing it to flexibly capture multi-scale features. The model refines cross-entropy by adjusting the importance of polynomial bases, thereby improving the accuracy of retinal OCT image classification.	D4	Acc.	Prec.	Recall
			CNV	1.0000	0.9960	1.0000
			DME	0.9960	1.0000	0.9960
			Drusen	1.0000	0.9960	1.0000
			Normal	0.9960	1.0000	0.9960
			Ave.	0.9980	0.9980	0.9980
			D6	Acc.	Prec.	Recall
			AMD	1.0000	1.0000	1.0000
			CNV	0.9489	0.9389	0.9571
			CSR	1.0000	1.0000	1.0000
			DME	0.9439	0.9512	0.9457
			DR	1.0000	0.9972	1.0000
			Drusen	0.9200	0.9580	0.9114
			MH	1.0000	1.0000	0.9971
			Normal	0.9563	0.9254	0.9571
			Ave.	0.9711	0.9713	0.9711
[52]	Focused Attention Transformer	Focused Attention employs iterative conditional patch resampling to produce interpretable predictions through high-resolution attribution maps.	D4*	Acc. (%)	Spec. (%)	Recall (%)
			T2T-ViT_14	94.40	98.13	94.40
			T2T-ViT_19	93.20	97.73	93.20
			T2T-ViT_24	93.40	97.80	93.40
[53]	ViT with Logit Loss Function	Captures global features via self-attention mechanism, reducing reliance on local texture features. Adjusts classifier’s logit weights and modifies them to a logit cross-entropy function with L2 regularization as a loss function.	D7*	Acc (%)	Sen. (%)	Spec. (%)
			Early DME	90.87	87.03	93.02
			Advanced DME	89.96	88.18	90.72
			Severe DME	94.42	63.39	98.4
			maculopathy	95.13	89.42	96.66
			OA: 87.3%			
[54]	Model-Based ViT (MBT-ViT), Model-Based ViT (MBT-SwinT), Multi-Scale Model-Based ViT (MBT-ViT)	Approximate sparse representation MBT utilizes ViT Swin ViT and multi-scale ViT for OCT video classification, then estimates key features before performing data classification.	D4	Acc.	Recall	
			MBT ViT	0.8241	0.8138	
			MBT SwinT	0.8276	0.8172	
			MBT_MViT	0.9683	0.9667	
[55]	Structure-Oriented Transformer (SoT)	SoT employs guidance mechanism that acts as a filter to emphasize the entire retinal structure. Utilizes vote classifier, which optimizes the utilization of all output tokens to generate the final grading results.		B-acc	Sen	Spe
			^D1^SoT	0.9935	0.9925	0.9955
			^D5^ SoT	0.9935	0.9925	0.9955
[56]	OCT Multihead Self-Attention (OMHSA)	OMHSA enhances self-attention mechanism by incorporating local information extraction, where a network architecture, called OCTFormer, is built by repeatedly stacking convolutional layers and OMHSA blocks at each stage.	D4	ACC	Prec.	Sen.
			OCT Former-T	94.36	94.75	94.37
			OCT Former-S	96.67	96.78	96.68
			OCT Former-B	97.42	97.47	97.43
[57]	Squeeze Vision transformer (S-ViT)	SViT combines SqueezeNet and ViT to capture local and global features, which enables more precise classification while maintaining lower computational complexity.	D5 Acc.: 0.9990, Sen.: 0.9990, Prec.: 1.000			
[14]	Deep Relation Transformer (DRT)	DRT integrates both OCT and Vision Field (VF) data, where this model incorporates a deep reasoning mechanism to identify pairwise relationships between OCT and VF.	^D10^Ablation Study			
			Back-bone	Acc (%)	Sen (%)	Spec (%)
			Light ResNet	**88.3 ± 1.0**	**93.7 ± 3.5**	**82.4 ± 4.1**
			ResNet-18	87.6 ± 2.3	93.1 ± 2.4	82.1 ± 4.3
			ResNet-34	87.2 ± 1.6	90.4 ± 5.0	83.9 ± 3.6
[58]	Conv-ViT–inception V3 and ResNet50	Integrates Inception-V3 and ResNet-50 to capture texture information by evaluating the relationships between nearby pixels. A Vision Transformer processes shape-based features by analyzing correlations between distant pixels.	D4	Feature Level Concatenation	Decision Level Conc.	
			Acc.	94.46%	87.38%	
			Prec.	0.94	0.87	
			Recall	0.94	0.86	
			F1 Score	0.94	0.86	
[59]	Multi-contrast Network	ViT cross-modal multi-contrast network utilizes multi-contrast learning to extract features. Then, a channel fusion head aggregates across different modalities.	D4	Acc (%)	SE (%)	SP (%)
			Normal	99.5	99.38	100
			CNV	100	100	100
			DR	99.5	100	99.42
			AMD	100	100	100
			All	99.75	99.84	99.85
[60]	Swin Transformer V2 with Poly Loss Function	Swin Transformer V2 leverages self-attention within local windows while using a PolyLoss function.	D4	Acc.	Recall	Spec.
			CNV	0.999	1.00	0.996
			DME	0.999	1.00	1.00
			DRUSEN	1.00	1.00	1.00
			NORMAL	1.00	1.00	1.00
			D6	Acc.	Recall	Spec.
			AMD	1.00	1.00	1.00
			CNV	0.989	0.949	0.995
			CSR	1.00	1.00	1.00
			DME	0.992	0.977	0.995
			DR	1.00	1.00	1.00
			DRUSEN	0.988	0.934	0.995
			MH	1.00	1.00	1.00
			NORMAL	0.991	0.98	0.992
[61]	Lesion-Localization Convolution Transformer (LLCT)	LLCT combines CNN-extracted feature maps with a self-attention network to capture both local and global image context. The model uses backpropagation to adjust weights, enhancing lesion detection by integrating global features from forward propagation.	D4	Acc (%)	Sens (%)	Spec. (%)
			CNV	98.1 ± 1.9	99.4 ± 0.3	97.6 ± 2.7
			DME	99.6 ± 0.2	99.6 ± 0.0	99.5 ± 0.3
			Drusen	98.1 ± 2.3	92.8 ± 8.5	99.9 ± 0.2
			Norm	99.6 ± 0.6	98.8 ± 1.7	99.9 ± 0.2
[62]	Stitched MedViTs	Stitching approach combines two MedViT models to find an optimal architecture. This method inserts a linear layer between pairs of stitchable layers, with each layer selected from one of the input models, creating a candidate model in the search space.	D4	Spec.	Acc.	
			micro MedViT	0.928 ± 0.002	0.828 ± 0.007	
			tiny MedViT	0.933 ± 0.002	0.841 ± 0.007	
			micro MedViT	0.987 ± 0.001	0.977 ± 0.002	
			tiny MedViT	0.986 ± 0.002	0.977 ± 0.004	
[63]	Bidirectional Gated Recurrent Unit (GRU)	Combines a pre-trained Vision Transformer for slice-wise feature extraction with a bidirectional GRU to capture inter-slice spatial dependencies, enabling analysis of both local details and global structural integrity.	D4	ACC	SEN	SPE
			ResNet34 + GRU	87.39 (± 1.73)	92.03	72.86
			ViT-large + GRU	90.27 (± 1.44)	94.25	78.18

^D1^D1, ^D5^D5, ^D10^D10. OA: Overall Accuracy, OS: Overall Sensitivity, OP: Overall Precision, OF1: Overall F1; binary classifications with AMD and normal classes; NB: naïve Bayes; RF: Random Forest; SVM: support vector machine.

**Table 9 bioengineering-12-00914-t009:** Adversarial samples and techniques employed in medical imaging.

Ref	Adversarial Samples Introduced	Modality	Technique Employed	Future Direction
[19]	Gaussian distributed noise with various noise levels	OCT images	MKW-LBP local descriptor with SVM and Random Forest classifiers.	Analyze how the application of Gaussian noise to OCT images affects classification performance metrics across various disease classes, including CNV, CSR, macular hole, and others.
[83]	Pepper noise with various noise densities	Skin cancer images	OS-LBP codes skin cancer images and is used to train CNN models. Trained models are employed for identifying potential skin cancer areas and mitigating the effects of image degradation.	Examine the effects of Salt and Pepper noise applied to OCT images across various diseases.
[84]	Contrast degradation	Endoscopic Images	Encodes WCE images using CQ-MPP and is used to train CNN models. Experts are employed for identifying areas of lesions and mitigating the effects of contrast degradation.	Adjusting the brightness and contrast of OCT images and observing their effects on performance.
[85]	Fast Gradient Sign Method (FGSM)	Skin cancer images, MRI	Adversarial training using inception for skin cancer classification and brain tumor segmentations.	Apply FGSM on OCT images on other deep learning networks such as ViT, Swin Transformer, and other DL techniques. Observe the effects on their performance metrics.
[86]	FGSM Perturbations, Basic Iterative Method (BIM), Projected Gradient Descent (PGD), Carlini and Wagner (CW) Attack	Eye fundus, lung X-rays, skin cancer images	KD models normal samples within the same class as densely clustered in a data manifold, whereas adversarial samples are distributed more sparsely outside the data manifold. LID is a metric used to describe the dimensional properties of adversarial subspaces in the vicinity of adversarial examples.	Apply BIM, PGD, and CW on OCT images on other deep learning networks such as ViT, Swin Transformer, and other DL techniques. Observe the effects on their performance metrics. Modify DL technique to mitigate the adversarial effects.
[90]	Frequency constraint-based adversarial attack	3D-CT, a 2D chest X-Ray image dataset, a 2D breast ultrasound dataset, and a 2D thyroid ultrasound	A perturbation constraint, known as the low-frequency constraint, is introduced to limit perturbations to the imperceptible high-frequency components of objects, thereby preserving the similarity between the adversarial and original examples.	Apply frequency constraint-based attacks to OCT images in the frequency domain and observe their impact on the images in the spatial domain.
[91]	Model Ensemble Feature Fusion (MEFF)	Fundoscopy, chest X-ray, dermoscopy	The MEFF approach is designed to mitigate adversarial attacks in medical image applications by combining features extracted from multiple deep learning models and training machine learning classifiers using these fused features.	Applying MEFF on two or more DL techniques to observe how well it mitigates classification errors.
[92]	Multi-View Learning	Natural RGB images	A multi-view classification method with an adversarial sample uses the evidential dissonance measure in subjective logic to evaluate the quality of data views when subjected to adversarial attacks.	Generate multi-view representations of OCT images and assess their impact on classification performance.
[93]	Medical morphological knowledge-guided	Lung CT scans	This approach trains a surrogate model with an augmented dataset using guided filtering to capture the model’s attention, followed by a gradient normalization-based prior knowledge injection module to transfer this attention to the main classifier.	Train the model on OCT images using a morphology-guided approach and evaluate its effectiveness in reducing classification errors.

## Data Availability

Please see citations of datasets.

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
