# Peer review of "Comprehensive Survey of OCT-Based Disorders Diagnosis: From Feature Extraction Methods to Robust Security Frameworks"

_bioengineering, 2025, doi:10.3390/bioengineering12090914_

Round 1
Reviewer 1 Report
Comments and Suggestions for Authors
This manuscript provides a broad and well-organized review of OCT-based methods for ocular disease diagnosis. It covers both hand-crafted and deep learning feature extraction techniques and includes a detailed summary of publicly available OCT datasets. The authors also attempt to explore emerging areas such as adversarial robustness and LLMs, which are timely and relevant.
While the review is ambitious in scope and contains useful reference material, it currently lacks the analytical depth, theoretical synthesis, and critical comparison expected of a high-quality survey article. Much of the content is descriptive rather than evaluative, and several important methodological and clinical considerations are not sufficiently addressed. The manuscript would benefit from a more structured conceptual framework, deeper technical analysis, and significant improvements in language clarity. There are several major concerns:
- The review lacks a unified comparative framework. While many methods are discussed, there is no coherent structure to compare them based on task relevance, data dependency, interpretability, or robustness.
- The discussion of adversarial attacks and model security is superficial. These sections mention emerging challenges but do not provide technical depth, recent citations, or actionable insights.
-
Several claims of novelty are overstated. The assertion that no prior surveys compare hand-crafted and deep features is inaccurate and should be revised to reflect existing literature.
-
The coverage of OCT datasets is comprehensive, but there is minimal discussion of dataset bias, imaging heterogeneity, or domain shift.
-
The writing style includes colloquial expressions, repetitions, and awkward phrasing. The overall presentation would benefit from professional language polishing and more concise formulation.
Reviewer 2 Report
Comments and Suggestions for Authors
The paper provides a Comprehensive Survey of the OCT-Based Disorders Diagnosis: From Feature Extraction Methods to Robust Security Frame- works. After reviewing the paper, I have the following comments.
[1] In figure 1, individual labels should be given to each photo for a better presentation.
[2] In section 1.1, it may be better to present the summary of the literature survey in forms of table in which reference and relative merits are summarized. This may be useful for other researchers.
[3] In table 2, an extra column could be included to summarized the characteristics of the dataset mentioned in Table 2.
[4] A table under figure 3 could be added to summarize the relative merits for different CNN structure and summary to describe the situtation that a particular CNN may give beter performance.
[5] The fonts and label in Figure 3 should be enlarged for better presentation.
[6] For the comparative analysis in figure 4, fonts and label in Figure 4 should be enlarged for better presentation.
[7] For the comparative analysis and results, the presentation of results could be in form of data bar chart to show the relative advantages of different method.
[8] The authors could summarize their view on future work in form of a table with some reference for the proposed research directions. Such information will be useful for other researchers.
[9] For figure 9, individual labels could be added in each photo for better presentation.
[10] The conclusion in section 8 is quite brief, the authors could extend this section to describe more about the original contributions of the paper.
